# Burden of disease and risk factors for mortality amongst hospitalized newborns in Nigeria and Kenya

Helen M. Nabwera[1,2]*, Dingmei Wang[3], Olukemi O. Tongo[4], Pauline E. A. Andang'o[5], Isa Abdulkadir[6], Chinyere V. Ezeaka[7], Beatrice N. Ezenwa[7], Iretiola B. Fajolu[7], Zainab O. Imam[8], Martha K. Mwangome[9], Dominic D. Umoru[10], Abimbola E. Akindolire[4], Walter Otieno[5,11], Grace M. Nalwa[5,11], Alison W. Talbert[9], Ismaela Abubakar[1], Nicholas D. Embleton[12,13], Stephen J. Allen[1,2], on behalf of the Neonatal Nutrition Network (NeoNuNet)¶

1 Liverpool School of Tropical Medicine, Liverpool, United Kingdom, 2 Alder Hey Children's Hospital NHS Trust, Liverpool, United Kingdom, 3 Children's Hospital of Fudan University, Minhang District, Shanghai, China, 4 University College Hospital, Ibadan, Nigeria, 5 Maseno University, Maseno, Kenya, 6 Ahmadu Bello University Teaching Hospital, Shika, Zaria, Nigeria, 7 Lagos University Teaching Hospital, Idi-Araba, Lagos, Nigeria, 8 Massey Street Children's Hospital, Lagos, Nigeria, 9 KEMRI-Wellcome Trust Research Programme, Kilifi, Kenya, 10 Maitama District Hospital, Maitama, Abuja, Nigeria, 11 Jaramogi Oginga Odinga Teaching and Referral Hospital, Jomo Kenyatta Highway Kaloleni Kisumu KE, Central, Kenya, 12 Newcastle University, Newcastle upon Tyne, United Kingdom, 13 The Newcastle upon Tyne Hospitals NHS Foundation Trust, High Heaton, Newcastle upon Tyne, United Kingdom

¶ Membership of the author group is listed in the Acknowledgments.
* helen.nabwera@lstmed.ac.uk

## Abstract

### Objective

To describe the patient population, priority diseases and outcomes in newborns admitted <48 hours old to neonatal units in both Kenya and Nigeria.

### Study design

In a network of seven secondary and tertiary level neonatal units in Nigeria and Kenya, we captured anonymised data on all admissions <48 hours of age over a 6-month period.

### Results

2280 newborns were admitted. Mean birthweight was 2.3 kg (SD 0.9); 57.0% (1214/2128) infants were low birthweight (LBW; <2.5kg) and 22.6% (480/2128) were very LBW (VLBW; <1.5 kg). Median gestation was 36 weeks (interquartile range 32, 39) and 21.6% (483/2236) infants were very preterm (gestation <32 weeks). The most common morbidities were jaundice (987/2262, 43.6%), suspected sepsis (955/2280, 41.9%), respiratory conditions (817/2280, 35.8%) and birth asphyxia (547/2280, 24.0%). 18.7% (423/2262) newborns died; mortality was very high amongst VLBW (222/472, 47%) and very preterm infants (197/483, 40.8%). Factors independently associated with mortality were gestation <28 weeks (adjusted odds ratio 11.58; 95% confidence interval 4.73–28.39), VLBW (6.92; 4.06–11.79), congenital anomaly (4.93; 2.42–10.05), abdominal condition (2.86; 1.40–5.83), birth

**Funding:** This project was completed as part of the Neonatal Nutrition Network, funded by a grant from the MRC Confidence in Global Nutrition and Health Research scheme, awarded to SJA (grant reference MC_PC_MR/R019789/1). The funders had no role in study design, data collection and analysis, decision to publish, or preparation of the manuscript.

**Competing interests:** The authors have declared that no competing interests exist.

asphyxia (2.44; 1.52–3.92), respiratory condition (1.46; 1.08–2.28) and maternal antibiotics within 24 hours before or after birth (1.91; 1.28–2.85). Mortality was reduced if mothers received a partial (0.51; 0.28–0.93) or full treatment course (0.44; 0.21–0.92) of dexamethasone before preterm delivery.

## Conclusion

Greater efforts are needed to address the very high burden of illnesses and mortality in hospitalized newborns in sub-Saharan Africa. Interventions need to address priority issues during pregnancy and delivery as well as in the newborn.

## Introduction

Globally, 2.5 million infants died in the neonatal period (before age 28 days) in 2018. Neonatal deaths accounted for 47% of all under 5 deaths and this proportion is increasing [1]. The majority of neonatal deaths occur in the first week of life and are preventable with equitable access to adequate, evidence-based maternal and newborn health care [2]. Sub-Saharan Africa (SSA) bears a high burden of adverse neonatal outcomes and is the region where the least progress has been made in addressing neonatal morbidity and mortality [1, 3]. The third Sustainable Development Goal emphasises the need to end preventable newborn deaths with a target for all countries to reduce the neonatal mortality rate (NMR) to 12 per 1000 live births or lower by 2030 [4].

Globally, preterm (gestation <37 weeks) and low birthweight (LBW; <2.5 kg) infants have an increased risk of mortality in the neonatal and post neonatal periods [5, 6]. Over 97% of these infants are born in resource-limited settings [5, 7–9]. Indeed, complications of preterm birth are now the leading cause of under 5 deaths accounting for 18% [1, 10]. Intrapartum-related events and neonatal sepsis are other major causes of under 5 mortality [1].

Nigeria and Kenya are both ranked as lower-middle income countries by the United Nations [11]. The NMR has halved in both countries over the past five decades but they remain amongst the countries with the highest NMR in SSA, with the estimated NMR in 2018 in Nigeria of 39 per 1000 live births and in Kenya 19.6 per 1,000 live births [12, 13]. Both countries are unlikely to meet the SDG target at their current rate of progress [3].

Reliable estimates of underlying causes of newborn morbidity and mortality are a pre-requisite for evidence-based policy-making, advocacy and priority setting for future research [14]. Unfortunately, in many SSA countries including Nigeria and Kenya, health information systems are inadequate [15, 16]. Routine clinical data on common, serious neonatal conditions such as preterm birth, low birthweight (LBW), birth asphyxia, sepsis and respiratory disorders in hospitalised infants are sparse [17–19]. This lack of routine patient data, alongside the limited engagement of health professionals managing these infants in research, inhibits the development of context relevant strategies and novel interventions to improve outcomes.

We established the Neonatal Nutrition Network (NeoNuNet) consisting of five neonatal units (NNUs) in Nigeria and two in Kenya to share expertise and experience to improve clinical outcomes across the Network and provide a platform for the development and evaluation of nutrition and other key interventions for the most at risk preterm/LBW infants. In the first phase a shared database of anonymised routine clinical data was therefore established to describe the patient population, priority diseases and outcomes in newborns admitted <48 hours old to inform the development of context-relevant interventions in these SSA NNUs.

## Methods

### Ethics approval

The study was approved by the Research and Ethics Committee at the Liverpool School of Tropical Medicine (protocol number:18–0210), The Lagos University Teaching Hospital Health Research Ethics Committee (protocol number: AMD/DCST/HREC/APP/2514), The Kenya Medical Research Institute-Scientific and Ethics Review Unit (protocol number: KEMRI/SERU/CGMR-C/120/3740) and the research and ethics committees at The Jaramogi Oginga Odinga Teaching and Referral Hospital (protocol number: ERC.IB/VOL.1/510), University College Hospital Ibadan (protocol number: UI/EC/18/0446), Massey Street Children's Hospital (protocol number: LSHSC/2222/VOL.VI[B]/185), Ahmadu Bello University Teaching Hospital (ABUTH/HZ/HREC/D37/2018), and Maitama District Hospital (protocol number: FHREC/2018/01/108/19-09-18).

### Study setting

The study was conducted in five NNUs in Nigeria of which four provide tertiary level care (University College Hospital, Ibadan; Lagos University Teaching Hospital, Massey Children's Hospital, Lagos; Ahmadu Bello University Teaching Hospital, Zaria) and one secondary level neonatal care (Maitama District Hospital, Abuja) and two in Kenya: Jaramogi Oginga Odinga Teaching and Referral Hospital, Kisumu providing tertiary level and Kilifi County Referral Hospital secondary level neonatal care. In Nigeria, the Nigeria Society of Neonatal Medicine led in the selection of these facilities and aimed to incorporate neonatal units from both the northern and southern parts of the country. In Kenya, the facilities were chosen based on previous collaborative partnerships aiming to include neonatal units that provide different levels of care i.e. tertiary and district level in different regions of the country. The basis of this selection process was prior research and clinical training collaborative partnerships between the Liverpool School of Tropical Medicine (LSTM) co-investigators and clinical researchers in Nigeria and Kenya. All neonatal units admitted both inborn and outborn infants <28 days of age. All neonatal units except one in Kenya hadseparate rooms/wards for admitting inborn and outborn neonates. The tertiary level units typically had 2–4 consultant neonatologists or paediatricians who supervised resident doctors/registrars, intern house officers, clinical officers and a team of 1–3 nurses per shift. In all the units, the neonates were admitted by intern house officers, medical officers or clinical officers and were reviewed by a consultant neontologist or paediatrician daily during their admission. The bed capacity ranged from 24–80 but occupancy often exceeded 100%. The district level NNUs had 1–3 consultant paediatricians who worked with medical officers, clinical officers and a team of 2–3 nurses per shift. The bed capacity ranged from 12–27. The NNUs provide care according to institutional neonatal protocols in Nigeria and the Kenya national paediatric protocol [20]. All the NNUs had access to oxygen, pulse oximetry and phototherapy but these were often limited in availability and, therefore, reserved for the sickest newborns. All the tertiary level NNUs and one of the district level NNUs used non-invasive ventilation (i.e. continuous positive airway pressure), but none used endotracheal ventilation.

### Study design, study population and data collection

This was a multi-centre, prospective, observational study. During network meetings held before data collection, we established a standardised case report form (CRF) and an anonymised demographic/clinical database. Clinical criteria, laboratory analyses and imaging currently used for the diagnosis of common neonatal morbidities were reviewed and additional CRFs developed to capture episodes of suspected sepsis, respiratory problems, abdominal

conditions and birth asphyxia diagnosed according to current clinical practice in each NNU. The CRFs are available from https://www.lstmed.ac.uk/nnu. All newborns aged <48 hours admitted to each NNU over a 6-month period between August 2018 and May 2019 were included in the study and had CRFs completed with the period of data collection determined by timing of ethics approval. Infants who were ≥ 48 hours of age at admission were excluded because we wanted to optimize recall of information on feeding practices from birth. In addition, infants aged 48 hours and above were generally admitted to paediatric wards rather than neonatal units. Details of maternal demography, socioeconomic status, health and the current pregnancy were recorded from ante-natal records. Details of labour and delivery were collected from hospital records. Paper CRFs were kept as part of infant case records or stored separately and updated during NNU admission. Details of clinical criteria used to diagnose infant conditions were recorded in separate forms, the analyses of which are provided in a separate manuscript that is in preparation.

### Data management

Data clerks in each NNU entered data into a REDcap database (http://www.project-redcap.org/ ), that was hosted by the Liverpool School of Tropical Medicine (LSTM). Infants were identified by a unique study number only and no personal identifiers were recorded in the database.

### Data analysis

Categorical variables were presented as frequencies and percentages. Normally distributed variables were reported using means and standard deviations (SDs) and median and interquartile ranges (IQRs) were used for non-normally distributed variables. We analysed variables according to country and level of care to provide some insights into the variability between NNUs. When evaluating differences between individual NNUs, country and level of care, we considered the clinical relevance of differences in variables as well as statistical significance. Except for five variables with ≥10% missing data (maternal HIV status, hepatitis B, syphilis, gestational diabetes, and infant length of admission), the average percentage of missing data for variables ranged from 0–6.3%. The five variables with a high percentage of missing data were not included in the multivariable logistic regression analysis. Univariate and multivariable logistic regression analyses identified factors associated with mortality with data imported into Stata version 15.0 (Stata Corp). Multivariable logistic regression odds ratio plot was performed by R V3.5.2. Kaplan-Meier survival analysis was used to estimate the independent effects of gestation and birthweight on neonatal mortality. Mothers/infants with missing data for a variable were not included in the analyses.

Information about the collection of anonymised data was displayed in each NNU; no parents chose to opt-out of the study. At one of the Network sites, parents provided written informed consent; 82 parents declined consent at this site.

### Role of funding source

The funders of the study had no role in study design, data collection, data analysis, data interpretation or writing of the report.

## Results

Across the Network, anonymised data regarding 2280 infants admitted before age 48 hours were collected. Almost all variables for maternal demography and health (Table 1), labour and delivery (Table 2) and newborns (Tables 3 and 4) differed significantly amongst the NNUs.

**Table 1. Maternal demographics and health.**

| Variable | TOTAL | NIGERIA | | | | | KENYA | | P value |
|---|---|---|---|---|---|---|---|---|---|
| | N = 2280 | 1 | 2 | 3 | 4 | 5 | 6 | 7 | |
| | | N = 100 | N = 488 | N = 226 | N = 382 | N = 208 | N = 292 | N = 584 | |
| Maternal age in years, n | 2215 | 100 | 487 | 226 | 382 | 207 | 289 | 524 | |
| mean (SD) | 28.7 (6.2) | 30.8 (4.6) | 30.6 (5.6) | 31.0 (5.4) | 31.2 (5.8) | 28.7 (6.6) | 25.7 (6.2) | 25.6 (5.5) | |
| <18, n (%) | 48 (2.2) | 0 (0) | 3 (0.6) | 0 (0) | 0 (0) | 4 (1.9) | 18 (6.3) | 23 (4.4) | <0.001 |
| 18–29 n (%) | 1164 (52.5) | 42 (42.0) | 201 (41.3) | 82 (36.3) | 154 (40.3) | 114 (55.1) | 198 (68.5) | 373 (71.2) | |
| > = 30, n (%) | 1003 (45.3) | 58 (58.0) | 283 (58.1) | 144 (63.7) | 228 (59.7) | 89 (43.0) | 73 (25.3) | 128 (24.4) | |
| Maternal education, n | 2146 | 100 | 467 | 226 | 382 | 206 | 287 | 478 | |
| Did not complete primary education, n (%) | 297 (13.9) | 2 (2.0) | 14 (3.0) | 0 (0) | 19 (5.0) | 53 (25.7) | 126 (43.9) | 83 (17.3) | <0.001 |
| Completed only primary school, n (%) | 621 (28.9) | 12 (12.0) | 38 (8.1) | 69 (30.5) | 208 (54.4) | 33 (16.0) | 92 (32.1) | 169 (35.4) | |
| Completed secondary school, n (%) | 533 (24.8) | 15 (15.0) | 165 (35.3) | 43 (19.0) | 99 (25.9) | 41 (19.9) | 30 (10.4) | 140 (29.3) | |
| Completed tertiary level education, n (%) | 695 (36.4) | 71 (71.0) | 250 (53.5) | 114 (50.4) | 56 (14.7) | 79 (38.4) | 39 (13.6) | 86 (18.0) | |
| Maternal occupation, n | 2155 | 100 | 468 | 226 | 382 | 207 | 288 | 484 | |
| Unemployed or housewife, n (%) | 809 (37.5) | 35 (35.0) | 69 (14.8) | 50 (22.1) | 49 (12.8) | 123 (59.4) | 187 (64.9) | 296 (61.2) | <0.001 |
| Petty trader/ labourer, n (%) | 680 (31.5) | 21 (21.0) | 112 (23.9) | 79 (35.0) | 247 (64.7) | 37 (17.9) | 61 (21.2) | 123 (25.4) | |
| Junior schoolteachers/drivers, n (%) | 320 (14.9) | 7 (7.0) | 169 (36.1) | 30 (13.3) | 43 (11.3) | 15 (7.2) | 23 (8.0) | 33 (6.8) | |
| Intermediate public servant/senior schoolteachers, n (%) | 187 (8.7) | 13 (13.0) | 66 (14.1) | 44 (19.5) | 20 (5.2) | 14 (6.8) | 13 (4.5) | 17 (3.5) | |
| Senior public servant/ professionals/ large scale traders, n (%) | 159 (7.4) | 24 (24.0) | 52 (11.1) | 23(10.2) | 23 (6.0) | 18 (8.7) | 4 (1.4) | 15 (3.1) | |
| Marital status, n | 2188 | 100 | 464 | 226 | 382 | 206 | 279 | 531 | |
| Single, n (%) | 175 (8.0) | 0 (0) | 22 (4.7) | 7 (3.1) | 4 (1.0) | 0 (0) | 32 (11.5) | 110 (20.7) | <0.001 |
| Married, n (%) | 2004 (91.6) | 100 (100.0) | 439 (94.6) | 219 (96.9) | 378 (99.0) | 206 (100.0) | 245 (87.8) | 417 (78.5) | |
| Divorced, n (%) | 9 (0.4) | 0 (0) | 3 (0.7) | 0 (0) | 0 (0) | 0 (0) | 2 (0.7) | 4 (0.8) | |
| Parity, n | 2216 | 100 | 486 | 226 | 382 | 207 | 292 | 523 | |
| 0, n (%) | 97 (4.4) | 0 | 94 (19.3) | 0 | 0 | 0 | 0 | 3 (0.6) | <0.001 |
| 1, n (%) | 624 (28.2) | 21 (21.0) | 125 (25.7) | 69 (30.5) | 88 (23.0) | 47 (22.7) | 97 (33.2) | 177 (33.8) | |
| >1, n (%) | 1495 (67.4) | 79 (79.0) | 267 (55.0) | 157 (69.5) | 294 (77.0) | 160 (77.3) | 195 (66.8) | 343 (65.6) | |
| Number of stillbirths, n | 2222 | 100 | 486 | 226 | 382 | 207 | 292 | 529 | |
| One, n (%) | 118 (5.3) | 6 (6.0) | 27 (5.6) | 16 (7.1) | 8 (2.1) | 13 (6.3) | 24 (8.2) | 24 (4.5) | <0.001 |
| Two or more, n (%) | 38 (1.7) | 2 (2.0) | 4 (0.8) | 13 (5.8) | 5 (1.3) | 2 (0.9) | 6 (2.1) | 6 (1.2) | |
| Antenatal clinic visits, n | 2140 | 98 | 481 | 225 | 380 | 199 | 290 | 467 | |
| Zero to three, n (%) | 824 (38.5) | 20 (20.4) | 115 (23.9) | 78 (34.7) | 263 (69.2) | 26 (13.1) | 134 (46.2) | 188 (40.3) | <0.001 |
| Four to seven, n (%) | 1073 (50.1) | 61 (62.2) | 234 (48.7) | 132 (58.7) | 96 (25.3) | 134 (67.3) | 149 (51.4) | 267 (57.2) | |
| Eight or more, n (%) | 243 (11.4) | 17 (17.4) | 132 (27.4) | 15 (6.6) | 21 (5.5) | 39 (19.6) | 7 (2.4) | 12 (2.5) | |
| Number of foetuses, n | 2267 | 100 | 483 | 226 | 382 | 207 | 292 | 577 | |

(*Continued*)

**Table 1.** (Continued)

| Variable | TOTAL | NIGERIA | | | | | KENYA | | P value |
|---|---|---|---|---|---|---|---|---|---|
| | N = 2280 | 1 | 2 | 3 | 4 | 5 | 6 | 7 | |
| | | N = 100 | N = 488 | N = 226 | N = 382 | N = 208 | N = 292 | N = 584 | |
| 1, n (%) | 1825 (80.5) | 90 (90.0) | 411 (85.1) | 179 (79.2) | 255 (66.8) | 176 (85.0) | 241 (82.5) | 473 (82.0) | <0.001 |
| 2, n (%) | 337 (14.9) | 7 (7.0) | 59 (12.2) | 36 (15.9) | 73 (19.1) | 25 (12.1) | 48 (16.4) | 89 (15.4) | |
| 3–5, n (%) | 105 (4.6) | 3 (3.0) | 13 (2.7) | 11 (4.9) | 54 (14.1) | 6 (2.9) | 3 (1.0) | 15 (2.6) | |
| HIV status, n | 2039 | 95 | 365 | 225 | 346 | 173 | 270 | 565 | |
| Positive, n (%) | 128 (6.3) | 2 (2.1) | 6 (1.6) | 7 (3.1) | 8 (2.3) | 2 (1.2) | 11 (4.1) | 92 (16.3) | <0.001 |
| Hepatitis B, n | 1079 | 91 | 242 | 224 | 345 | 159 | 1 | 17 | |
| Positive, n (%) | 25 (2.3) | 2 (2.2) | 11 (4.6) | 2 (0.9) | 7 (2.0) | 3 (1.9) | 0 (0) | 0 (0) | 0.24 |
| Syphilis, n | 1388 | 93 | 122 | 225 | 65 | 127 | 252 | 504 | |
| Positive, n (%) | 2 (0.1) | 0 (0) | 0 (0) | 0 (0) | 0 (0) | 0 (0) | 0 (0) | 2 (0.4) | 0.74 |
| Gestational diabetes, n | 1755 | 95 | 460 | 225 | 35 | 164 | 241 | 535 | |
| Yes, n (%) | 34 (1.9) | 1 (1.0) | 14 (3.0) | 11 (4.9) | 5 (14.3) | 0 (0) | 1 (0.4) | 2 (0.4) | <0.001 |
| Pregnancy induced hypertension, n | 2136 | 95 | 471 | 225 | 355 | 188 | 263 | 539 | |
| Yes, n (%) | 363 (17.0) | 15 (15.8) | 76 (16.1) | 51 (22.7) | 110 (31.0) | 41 (21.8) | 16 (6.1) | 54 (10.2) | <0.001 |
| Antepartum haemorrhage, n | 2184 | 97 | 478 | 226 | 379 | 194 | 268 | 542 | |
| Yes, n (%) | 161 (7.4) | 7 (7.2) | 44 (9.2) | 14 (6.2) | 41 (10.8) | 12 (6.2) | 19 (7.1) | 24 (4.4) | 0.010 |

## Maternal characteristics (Table 1)

Mean age was 28.7 years (SD 6.2) and only 48 mothers were <18 years. A minority of mothers had not completed primary education (297/2146, 13.9%), over a third were unemployed or housewives (i.e. did not work out of the house) (809/2255, 37.5%) and 8.0% (175/2189) were single. Two thirds of mothers were multiparous (1495/2216, 67.4%) and 156/2222 (7.0%) had had at least one stillbirth. Over one third of mothers (824/2140, 38.5%) had attended less than four antenatal clinics and almost 1 in 5 (442/2267; 19.5%) had a multiple pregnancy. Age, education levels and employment status were lower in the Kenyan than Nigerian NNUs and in secondary than tertiary level NNUs. The proportion of single mothers was higher in Kenya than Nigeria (S1 Table in S1 File).

Availability of data from ante-natal records varied markedly between NNUs. Overall, 128/2039 (6.3%) mothers were HIV positive with a higher proportion in the Kenya than Nigeria NNUs (S1 Table in S1 File). Hepatitis B prevalence was 2.4% in Nigeria but was rarely tested in Kenya (S1 Table in S1 File). Syphilis positivity was low (2/1388; 0.1%). Pregnancy induced hypertension (363/2136, 17.0%) and antepartum haemorrhage (161/2184, 7.4%) were common complications of pregnancy but gestational diabetes was less common (34/1755; 1.9%).

## Labour and delivery (Table 2)

Most births were facility-based (2128/2280; 93.3%). Risk factors for perinatal sepsis were common among mothers; 359/2151 (16.7%) had rupture of membranes ≥18 hours and 204/2159 (9.5%) had confirmed or suspected peripartum fever. More than half of mothers had received treatment with antibiotics within 24 hours before or after birth (1263/2147, 58.8%). 10.6% (115/1083) of mothers of preterm infants received four doses of antenatal dexamethasone. Almost half of deliveries were either vaginal assisted or by Caesarean section. Maternal mortality was 1.1% (24/2267).

**Table 2. Labour and delivery.**

| Variable | Total | NIGERIA | | | | | KENYA | | P value |
|---|---|---|---|---|---|---|---|---|---|
| | N = 2280 | 1 | 2 | 3 | 4 | 5 | 6 | 7 | |
| | | N = 100 | N = 488 | N = 226 | N = 382 | N = 208 | N = 292 | N = 584 | |
| **Place of delivery, n** | 2280 | 100 | 488 | 226 | 382 | 208 | 292 | 584 | |
| **Health facility, n (%)** | 2128 (93.3) | 98 (98.0) | 443 (90.8) | 213 (94.3) | 364 (95.3) | 179 (86.1) | 262 (89.7) | 569 (97.4) | <0.001 |
| Home, n (%) | 99 (4.4) | 1 (1.0) | 17 (3.5) | 9 (4.0) | 8 (2.1) | 29 (13.9) | 22 (7.5) | 13 (2.2) | |
| Other, n (%)* | 53 (2.3) | 1 (1.0) | 28 (5.7) | 4 (1.8) | 10 (2.6) | 0 (0) | 8 (2.7) | 2 (0.4) | |
| **Rupture of membranes ≥ 18 hours, n** | 2151 | 97 | 480 | 223 | 377 | 207 | 285 | 482 | |
| **Yes, n (%)** | 359 (16.7) | 16 (16.5) | 103 (21.5) | 51 (22.9) | 77 (20.4) | 33 (15.9) | 27 (9.5) | 52 (10.8) | <0.001 |
| **Maternal peripartum fever, n** | 2159 | 94 | 473 | 221 | 380 | 198 | 287 | 506 | |
| **Confirmed or suspected, n (%)** | 204 (9.5) | 21 (22.3) | 45 (9.5) | 13 (5.9) | 13 (3.4) | 40 (20.2) | 25 (8.7) | 47 (9.3) | <0.001 |
| **Mother treated with antibiotics within 24 hrs before/after birth, n** | 2147 | 90 | 423 | 217 | 375 | 193 | 281 | 568 | |
| **Yes, n (%)** | 1263 (58.8) | 69 (76.7) | 78 (18.4) | 136 (62.7) | 330 (88.0) | 75 (38.9) | 41 (14.6) | 534 (94.0) | <0.001 |
| **Mother <37 gestational weeks received antenatal dexamethasone, n** | 1083 | 29 | 237 | 129 | 302 | 62 | 109 | 215 | |
| **Full 4 doses, n (%)** | 115 (10.6) | 3 (10.3) | 50 (21.1) | 27 (20.9) | 13 (4.3) | 6 (9.7) | 0 (0) | 16 (7.4) | <0.001 |
| 1–3 doses, n (%) | 171 (15.8) | 7 (24.1) | 33 (13.9) | 42 (32.6) | 55 (18.2) | 24 (38.7) | 1 (0.9) | 9 (4.2) | |
| None, n (%) | 797 (73.6) | 19 (65.6) | 154 (65.0) | 60 (46.5) | 234 (77.5) | 32 (51.6) | 108 (99.1) | 190 (88.4) | |
| **Mode of delivery, n** | 2277 | 100 | 488 | 226 | 380 | 208 | 291 | 584 | |
| **CS, n (%)** | 1006 (44.2) | 56 (56.0) | 242 (49.6) | 140 (62.0) | 196 (51.6) | 89 (42.8) | 71 (24.4) | 212 (36.3) | <0.001 |
| **Vaginal assisted, n (%)** | 83 (3.6) | 3 (3.0) | 15 (3.1) | 9 (4.0) | 7 (1.8) | 11 (5.3) | 23 (7.9) | 15 (2.6) | |
| **Vaginal unassisted, n (%)** | 1188 (52.2) | 41 (41.0) | 231 (47.3) | 77 (34.0) | 177 (46.6) | 108 (51.9) | 197 (67.7) | 357 (61.1) | |
| **Maternal outcomes, n** | 2267 | 100 | 486 | 226 | 380 | 206 | 292 | 577 | |
| **Maternal death, n (%) **** | 24 (1.1) | 1 (1.0) | 13 (2.7) | 5 (2.2) | 5 (1.3) | 0 (0) | 0 (0) | 0 (0) | <0.001 |

*Top three: 20 at the mission home, 20 at traditional birth attendant home, and 7 delivered on the way to hospital

** Top four: postpartum haemorrhage (n = 7), eclampsia (n = 5), cardiac or pulmonary compromise (n = 4), HIV encephalopathy (n = 2).

## Newborn admission characteristics (Table 3)

Most infants were male (1292/2280, 56.7%). Mean birth weight was 2.3 kg (SD 0.9); over half of admissions were LBW (1214/2182, 55.6%) and about one in five (480/2182, 22.0%) were VLBW. Methods used to assess gestation varied markedly between NNUs with last menstrual period being the most common method (1587/2245, 70.7%). Median gestation was 36 weeks (IQR 32, 39). Over half of admissions were preterm (1172/2236, 52.4%) and about one in five were very preterm (gestation <32 weeks; 483/2236; 21.6%). Nearly half of all infants received prophylactic antibiotics (1113/2252, 49.4%; defined as infants with risk factors for sepsis but without clinical features of sepsis).

## Morbidity and mortality in infants (Table 4)

Jaundice was the commonest morbidity (987/2262, 43.6%; defined as an infant treated with phototherapy), with greater frequency reported from the NNUs in Nigeria. Other common

**Table 3. Newborns on admission.**

| Variable | Total | NIGERIA | | | | | KENYA | | P value |
|---|---|---|---|---|---|---|---|---|---|
| | N = 2280 | 1 | 2 | 3 | 4 | 5 | 6 | 7 | |
| | | N = 100 | N = 488 | N = 226 | N = 382 | N = 208 | N = 292 | N = 584 | |
| **Gender, n** | 2280 | 100 | 488 | 226 | 382 | 208 | 292 | 584 | |
| Male, n (%) | 1292 (56.7) | 60 (60.0) | 288 (59.0) | 125 (55.3) | 181 (47.4) | 120 (57.7) | 183 (62.7) | 335 (57.4) | 0.003 |
| **Birth weight, kg, n** | 2182 | 94 | 444 | 224 | 380 | 180 | 281 | 579 | |
| **Birth weight, mean (SD)** | 2.3 (0.9) | 2.7 (0.8) | 2.4 (0.9) | 2.3 (1.0) | 1.8 (0.7) | 2.5 (0.9) | 2.4 (0.9) | 2.5 (0.9) | <0.001 |
| <1, n (%) | 107 (4.9) | 0 (0) | 17 (3.8) | 16 (7.1) | 34 (9.0) | 3 (1.7) | 15 (5.4) | 22 (3.8) | |
| 1-<1.5, n (%) | 373 (17.1) | 11 (11.7) | 77 (17.3) | 51 (22.8) | 94 (24.7) | 19 (10.6) | 43 (15.3) | 78 (13.5) | |
| 1.5-<2.5, n (%) | 734 (33.6) | 20 (21.3) | 140 (31.5) | 63 (28.1) | 204 (53.7) | 70 (38.9) | 72 (25.6) | 165 (28.5) | |
| 2.5-<4, n (%) | 909 (41.7) | 57 (60.6) | 198 (44.5) | 83 (37.1) | 48 (12.6) | 80 (44.4) | 147 (52.3) | 296 (51.1) | |
| 4–5.5, n (%) | 59 (2.7) | 6 (6.4) | 12 (2.7) | 11 (4.9) | 0 (0) | 8 (4.4) | 4 (1.4) | 18 (3.1) | |
| **Gestation, weeks, n** | 2236 | 98 | 482 | 225 | 381 | 208 | 292 | 550 | |
| **median (IQR)** | 36 (32, 39) | 38 (34, 39) | 36 (32, 39) | 35 (30, 38) | 33 (30, 35) | 38 (36, 40) | 38 (33, 40) | 37 (33, 39) | |
| <28, n (%) | 119 (5.3) | 0 (0) | 24 (5.0) | 13 (5.8) | 40 (10.5) | 0 (0) | 14 (4.8) | 28 (5.1) | <0.001 |
| 28-<32, n (%) | 364 (16.3) | 8 (8.2) | 93 (19.3) | 55 (24.4) | 102 (26.8) | 9 (4.3) | 37 (12.7) | 60 (10.9) | |
| 32-<37, n (%) | 689 (30.8) | 27 (27.6) | 134 (27.8) | 69 (30.7) | 169 (44.4) | 57 (27.4) | 65 (22.3) | 168 (30.6) | |
| 37–42, n (%) | 981 (43.9) | 60 (61.2) | 220 (45.6) | 85 (37.8) | 68 (17.8) | 140 (67.3) | 141 (48.2) | 267 (48.5) | |
| 42–45, n (%) | 83 (3.7) | 3 (3.0) | 11 (2.3) | 3 (1.3) | 2 (0.5) | 2 (1.0) | 35 (12.0) | 27 (4.9) | |
| **Method used to assess gestation, n** | 2245 | 99 | 482 | 226 | 382 | 208 | 292 | 556 | |
| Ballard or other charts, n (%) | 425 (18.9) | 0 (0) | 18 (3.7) | 13 (5.8) | 5 (1.3) | 136 (65.4) | 239 (81.9) | 14 (2.5) | <0.001 |
| Early USS, n (%) | 233 (10.4) | 5 (5.1) | 143 (29.7) | 64 (28.3) | 6 (1.6) | 8 (3.9) | 1 (0.3) | 6 (1.1) | |
| Maternal last menstrual period, n (%) | 1587 (70.7) | 94 (94.9) | 321 (66.6) | 149 (65.9) | 371 (97.1) | 64 (30.8) | 52 (17.8) | 536 (96.4) | |
| **Admitted to NNU from, n** | 2280 | 100 | 488 | 226 | 382 | 208 | 292 | 584 | |
| Home, n (%) | 131 (5.8) | 13 (13.0) | 22 (4.5) | 10 (4.2) | 16 (4.2) | 43 (20.7) | 19 (6.5) | 8 (1.4) | <0.001 |
| Labour ward, n (%) | 910 (39.9) | 19 (19.0) | 234 (48.0) | 137 (60.6) | 111 (29.1) | 105 (50.5) | 91 (31.2) | 213 (36.5) | |
| Postnatal ward, n (%) | 202 (8.9) | 39 (39.0) | 15 (3.1) | 1 (0.4) | 1 (0.3) | 20 (9.6) | 61 (20.9) | 65 (11.1) | |
| Health facility, n (%) | 660 (28.9) | 13 (13.0) | 207 (42.4) | 78 (34.5) | 62 (16.2) | 31 (14.9) | 116 (39.7) | 153 (26.2) | |
| Other, n (%)* | 377 (16.5) | 16 (16.0) | 10 (2.0) | 0 (0) | 192 (50.3) | 9 (4.3) | 5 (1.7) | 145 (24.8) | |
| **Prophylactic antibiotics, n** | 2252 | 96 | 485 | 224 | 381 | 207 | 292 | 567 | |
| Yes, n (%) | 1113 (49.4) | 66 (68.8) | 269 (55.5) | 132 (58.9) | 370 (97.1) | 126 (60.9) | 33 (11.3) | 117 (20.6) | <0.001 |

*348 from theatre, and 22 from the clinics or wards; **Multiple choice.

conditions included suspected sepsis (955/2280, 41.9%), respiratory conditions (817/2280, 35.8%), birth asphyxia (547/2280, 24.0% in all infants but 463/1371, 33.8% among infants ≥35 weeks gestation) and congenital anomalies (128/2269, 5.6%). Of the 128 infants who were diagnosed with congenital abnormalities, 35 died during admission. These included: 10 with gastrointestinal abnormalities (1 with jejunal atresia, 1 with duodenal atresia, 1 with a gastric mass, 4 with gastroschisis, 3 with omphalocele), 6 with central nervous system abnormalities (2 with hydrocephalus, 2 with lumbar myelomeningocele, 1 with frontal encephalocele, 1 with microcephaly), 5 with cartilage and limb abnormalities (1 with achondroplasia, 1 with chrondrodysplasia, 1 congenital amputation of the right foot, 1 with genu recurvatum, 1 with congenital talipes), 4 with heart disease (both cyanotic and acyanotic), 2 with facial abnormalities (1 with cleft lip and palate, 1 with syngnathia) and 8 with other anomalies. An abdominal condition (including necrotising enterocolitis) occurred in 71/2280 (3.1%) infants with a frequency of 3.7% (45/1214) amongst LBW infants and 4.8% (23/483) amongst those with gestation <32 weeks.

**Table 4. Morbidity and mortality in infants.**

| Variable | TOTAL | NIGERIA | | | | | KENYA | | P value |
|---|---|---|---|---|---|---|---|---|---|
| | N = 2280 | 1 | 2 | 3 | 4 | 5 | 6 | 7 | |
| | | N = 100 | N = 488 | N = 226 | N = 382 | N = 208 | N = 292 | N = 584 | |
| **Congenital anomalies, n** | 2269 | 100 | 486 | 225 | 378 | 206 | 292 | 582 | |
| Yes, n (%) | 128 (5.6) | 1 (1.0) | 44 (9.1) | 13 (5.8) | 4 (1.1) | 7 (3.4) | 19 (6.5) | 40 (6.9) | <0.001 |
| **Congenital heart diseases, n** | 2234 | 99 | 473 | 226 | 367 | 203 | 292 | 574 | |
| Yes, n (%) | 56 (2.5) | 0 (0) | 37 (7.8) | 7 (3.1) | 0 (0) | 2 (1.0) | 4 (1.4) | 6 (1.1) | <0.001 |
| **Patent ductus arteriosus among infant birth weight <1.5kg, n** | 468 | 11 | 93 | 67 | 127 | 20 | 57 | 93 | |
| Yes, n (%) | 17 (3.6) | 0 (0) | 13 (14.0) | 1 (1.5) | 0 (0) | 1 (5.0) | 1 (1.8) | 1 (1.1) | <0.001 |
| **Received phototherapy, n** | 2262 | 100 | 484 | 226 | 380 | 204 | 291 | 577 | |
| Yes, n (%) | 987 (43.6) | 70 (70.0) | 254 (52.5) | 143 (63.3) | 281 (74.0) | 133 (65.2) | 47 (16.2) | 59 (10.2) | <0.001 |
| **Other common morbidities** | 2280 | 100 | 488 | 226 | 382 | 208 | 292 | 584 | |
| Asphyxia, n (%) | 547 (24.0) | 17 (17.0) | 168 (34.4) | 42 (18.6) | 38 (10.0) | 49 (23.4) | 53 (18.2) | 180 (30.8) | <0.001 |
| Respiratory conditions, n (%) | 817 (35.8) | 16 (16.0) | 198 (40.6) | 163 (72.1) | 21 (5.5) | 18 (8.7) | 85 (29.1) | 316 (54.1) | <0.001 |
| Abdominal condition, n (%) | 71 (3.1) | 6 (6.0) | 26 (5.3) | 18 (8.0) | 10 (2.6) | 3 (1.4) | 2 (0.7) | 6 (1.0) | <0.001 |
| Sepsis, n (%) | 955 (41.9) | 45 (45.0) | 247 (50.6) | 47 (20.8) | 47 (12.3) | 57 (27.4) | 141 (48.3) | 371 (63.5) | <0.001 |
| **Infant final outcome, n** | 2262 | 100 | 488 | 226 | 378 | 206 | 292 | 572 | |
| Absconded/discharge against medical, n (%) | 42 (1.9) | 2 (2.0) | 17 (3.5) | 2 (0.9) | 7 (1.9) | 5 (2.4) | 8 (2.7) | 1 (0.2) | <0.001 |
| Died, n (%) | 423 (18.7) | 4 (4.0) | 83 (17.0) | 36 (15.9) | 106 (28.0) | 36 (17.5) | 43 (14.7) | 115 (20.1) | |
| Transferred out, n (%) | 43 (1.9) | 3 (3.0) | 23 (4.7) | 1 (0.4) | 7 (1.8) | 0 (0) | 2 (0.7) | 7 (1.2) | |
| Discharged home with morbidities, n (%) | 55 (2.4) | 4 (4.0) | 18 (3.7) | 2 (0.9) | 1 (0.3) | 6 (2.9) | 14 (4.8) | 10 (1.8) | |
| Discharged home with no morbidities, n (%) | 1699 (75.1) | 87 (87.0) | 347 (71.1) | 185 (81.9) | 257 (68.0) | 159 (77.2) | 225 (77.1) | 439 (76.7) | |
| **Timing of mortality in infants, n** | 421 | 4 | 81 | 36 | 106 | 36 | 43 | 115 | |
| Age at time of death in days, median (IQR) | 2 (1, 5) | 2.5 (1, 6) | 3 (2, 6) | 2 (1, 4) | 2 (1, 4) | 5 (3, 7) | 2 (1, 5) | 1 (1, 3) | |
| < 7 days, n (%) | 335 (79.6) | 3 (75.0) | 57 (70.4) | 31 (86.1) | 90 (84.9) | 21 (58.3) | 33 (76.7) | 100 (87.0) | 0.003 |
| 7-<14 days, n (%) | 59 (14.0) | 1 (25.0) | 17 (21.0) | 4 (11.1) | 10 (9.4) | 13 (36.1) | 4 (9.3) | 10 (8.7) | |
| 15–30 days, n (%) | 23 (5.5) | 0 (0) | 6 (7.4) | 1 (2.8) | 4 (3.8) | 1 (2.8) | 6 (14.0) | 5 (4.3) | |
| >30 days, n (%) | 4 (0.9) | 0 (0) | 1 (1.2) | 0 (0) | 2 (1.9) | 1 (2.8) | 0 (0) | 0 (0) | |
| **Final outcome among infants birth weight <1.5kg, n** | 472 | 11 | 94 | 67 | 125 | 21 | 58 | 96 | |
| Absconded/discharge against medical, n (%) | 3 (0.6) | 0 (0) | 0 (0) | 1 (1.5) | 1 (0.8) | 0 (0) | 1 (1.7) | 0 (0) | 0.018 |
| Died, n (%) | 222 (47.0) | 1 (9.1) | 40 (42.6) | 24 (35.8) | 72 (57.6) | 14 (66.8) | 21 (36.2) | 50 (52.1) | |
| Discharged home with morbidities, n (%) | 4 (0.9) | 0 (0) | 2 (2.1) | 0 (0) | 0 (0) | 1 (4.8) | 1 (1.7) | 0 (0) | |
| Discharged home with no morbidities, n (%) | 237 (50.2) | 10 (90.9) | 49 (52.1) | 42 (62.7) | 50 (40.0) | 6 (28.6) | 35 (60.4) | 45 (46.9) | |
| Transferred out, n (%) | 6 (1.3) | 0 (0) | 3 (3.2) | 0 (0) | 2 (1.6) | 0 (0) | 0 (0) | 1 (1.0) | |
| **Final outcome among infants birth weight <1.0kg, n** | 104 | 0 | 17 | | | | 15 | 21 | |
| Absconded/discharge against medical, n (%) | 0 | 0 | 0 | | | | 0 | 0 | 0.023 |
| Died, n (%) | 81 (77.9) | 0 | 12 (70.6) | | | | 11 (73.3) | 18 (85.7) | |
| Discharged home with morbidities, n (%) | 0 | 0 | 0 | | | | 0 | 0 | |
| Discharged home with no morbidities, n (%) | 22 (21.2) | 0 | 5 (29.4) | | | | 4 (26.7) | 3 (14.3) | |
| Transferred out, n (%) | 1 (1.0) | 0 | 0 | | | | 0 | 0 | |
| **Timing of mortality in infants birth weight <1.5kg, n** | 221 | 1 | 39 | 24 | 72 | 14 | 21 | 50 | |

(*Continued*)

**Table 4.** (Continued)

| Variable | TOTAL | NIGERIA | | | | | KENYA | | P value |
|---|---|---|---|---|---|---|---|---|---|
| | N = 2280 | 1 | 2 | 3 | 4 | 5 | 6 | 7 | |
| | | N = 100 | N = 488 | N = 226 | N = 382 | N = 208 | N = 292 | N = 584 | |
| Age at time of death in days, median (IQR) | 2 (1, 6) | 3 (3, 3) | 3 (1, 7) | 2 (1, 3) | 2 (1, 5) | 5.5 (4, 8) | 3 (1, 6) | 1 (0, 5) | |
| < 7 days, n (%) | 165 (74.7) | 1 (100.0) | 26 (66.7) | 22 (91.7) | 58 (80.5) | 7 (50.0) | 13 (61.9) | 38 (76.0) | 0.042 |
| 7-<14 days, n (%) | 33 (14.9) | 0 (0) | 7 (17.9) | 2 (8.3) | 8 (11.1) | 6 (42.9) | 2 (9.5) | 8 (16.0) | |
| 15–30 days, n (%) | 19 (8.6) | 0 (0) | 5 (12.8) | 0 (0) | 4 (5.6) | 0 (0) | 6 (28.6) | 4 (8.0) | |
| >30 days, n (%) | 4 (1.8) | 0 (0) | 1 (2.6) | 0 (0) | 2 (2.8) | 1 (7.1) | 0 (0) | 0 (0) | |

Overall mortality during hospital admission was 18.7% (423/2262). Mortality was high amongst VLBW infants (222/472; 47.0%) and most extremely LBW (<1.0kg) infants died (81/104; 77.9%). Similarly, mortality was high amongst very preterm infants (197/483; 46.6%) and most extremely preterm (gestation<28 weeks) infants died (84/117; 71.8%). The majority of deaths were early neonatal deaths (335/421, 79.6%) with a median (IQR) age of death of 2 (1, 5) days (Table 4). Kaplan-Meier survival curves according to birth weight and gestation are shown in Figs 1 and 2 respectively. Survival was similar among infants with birth weight between 1.5-<2.5kg and those with weight ≥ 2.5 kg, but much lower in those with birth weight <1.5kg (log rank test P<0.001). The Kaplan-Meier survival probability estimates at 30 days were about 0.9 for infants with birthweight ≥1.5kg and 0.5 for VLBW infants. Similarly, survival was similar in infants with gestation between 32-<37 weeks and those gestation ≥37 weeks, but much lower in very preterm infants (log rank test P<0.001). The Kaplan-Meier survival probability estimates at 30 days were about 0.9 for infants with gestation ≥32 weeks and 0.5 for very preterm infants.

Of infants discharged home alive, the majority had no reported morbidities (1699/1754, 96.9%).

## Factors associated with mortality in infants

In univariate analysis, the factor with the highest odds ratio for mortality was extreme prematurity (gestation <28 weeks). Other factors significantly associated with increased mortality were twin delivery, antepartum haemorrhage, place of birth other than the health facility or home (which included in the car or a mission home), mother treated with antibiotics within 24 hrs before or after birth, female infant, VLBW, smaller head circumference, shorter length, gestation 28-<32 weeks, congenital anomaly, birth asphyxia, suspected sepsis, respiratory condition and abdominal condition. Factors protective against mortality included higher maternal education and occupation status, greater number of ANC visits and delivery by caesarean section (S2 Table in S1 File).

In multivariable logistic regression analysis, gestation <28 weeks remained the factor with the highest odds of death (adjusted odds ratio [AOR] 11.58, 95% confidence interval [CI] 4.73, 28.39). The following factors were also significantly associated with increased mortality: birth weight <1.5kg (AOR 6.92, 95%CI 4.06, 11.79), congenital anomaly (AOR 4.93, 95% CI 2.42, 10.05), infant having at least one abdominal condition (AOR 2.86, 95% CI 1.40, 5.83), birth asphyxia (AOR 2.44, 95%CI 1.52, 3.92), mother receiving treatment with antibiotics within 24 hours before or after birth (AOR 1.91, 95%CI 1.28, 2.85) and infant having a respiratory condition (AOR 1.46, 95%CI 1.08, 2.28). Mother receiving 1–3 doses (AOR 0.51, 95% CI 0.28, 0.93) or four doses (AOR 0.44, 95%CI 0.21, 0.92) of antenatal dexamethasone for preterm deliveries remained protective (Fig 3).

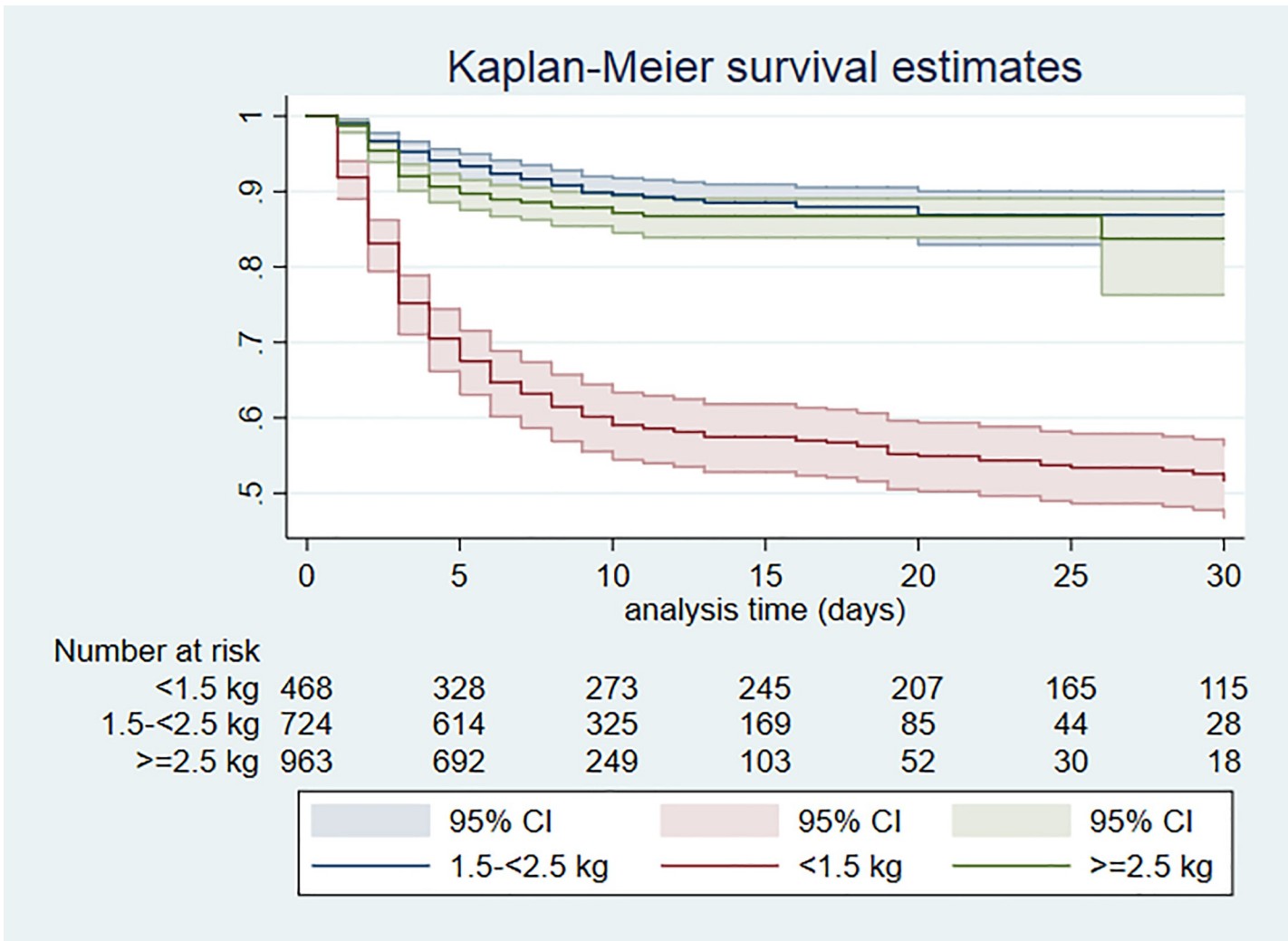

**Fig 1. Kaplan Meier survival by birthweight category; Day 0–30 of admission.**

Among infants with birth weight <1.5kg, factors associated with neonatal mortality were being extremely preterm (AOR 8.59, 95%CI 4.12, 17.93) and mother receiving treatment with antibiotics within 24 hours before or after birth (AOR 2.83, 95%CI 1.46, 5.50). Infant being female (AOR 0.53, 95%CI 0.31, 0.92) and mother receiving 1–3 doses (AOR 0.45, 95%CI 0.20, 0.98) or 4 doses (AOR 0.28, 95%CI 0.11, 0.70) of antenatal dexamethasone for preterm deliveries were protective (S3 Table in S1 File).

## Discussion

### Neonatal morbidity and mortality

We report a high burden of mortality in this prospective study of hospitalised newborns admitted to secondary and tertiary NNUs in Nigeria and Kenya with an overall mortality rate of 18.7%. Variability in mortality between NNUs was marked and ranged between 4.0–28.0%. Mortality in our study is consistent with prospective studies of admissions to six NNUs in public hospitals in Eastern Ethiopia (mortality of 20.0%) [21] and to a public teaching hospital in Addis Ababa, Ethiopia (23.1%) [22]. In retrospective studies, mortality ranged from 13–38%

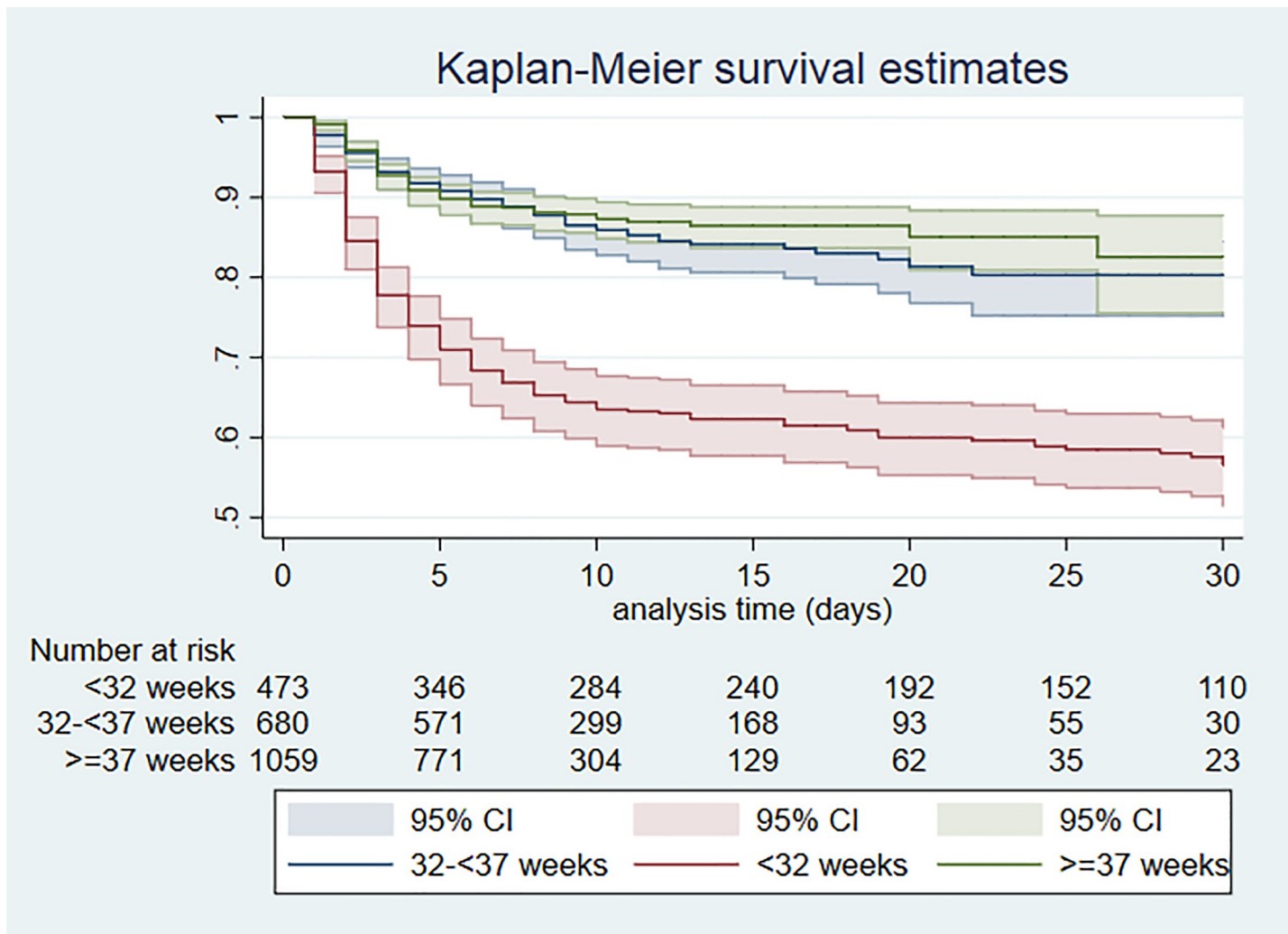

**Fig 2. Kaplan Meier survival by gestational age group; Day 0–30 of admission.**

in NNUs in Kenya [29, 30], 5.7–23.3% in Ethiopia [23, 24], 18.8% in Nigeria [25], 15.7% in Cameroon [26] and 8.2% in Eritrea [27].

Significant variation in case mix likely contributes to the marked variability in mortality between NNUs included in our study. Compared with the NNUs in Nigeria, maternal age, level of education and employment status were lower and HIV infection rates higher in Kenya. This suggests that the NNUs in Kenya served a more disadvantaged population which may be a reflection of the Free Maternity Service policy implemented in all public (government-run) hospitals in 2013 [28]. Interestingly, although this policy increased facility-based deliveries, recent data suggests that it has not significantly reduced maternal or neonatal mortality [29]. The case mix also varied between secondary and tertiary level NNUs with more women who were less educated and unemployed attending secondary level NNUs. Neonatal risk adjustment scores, using combinations of vulnerability (e.g. VLBW, prematurity), biological (e.g. common morbidities) and socioeconomic variables, will be important to compare outcomes between different NNUs, levels of care and regions and inform research and quality improvement initiatives, and are being developed specifically for low-resource settings [30].

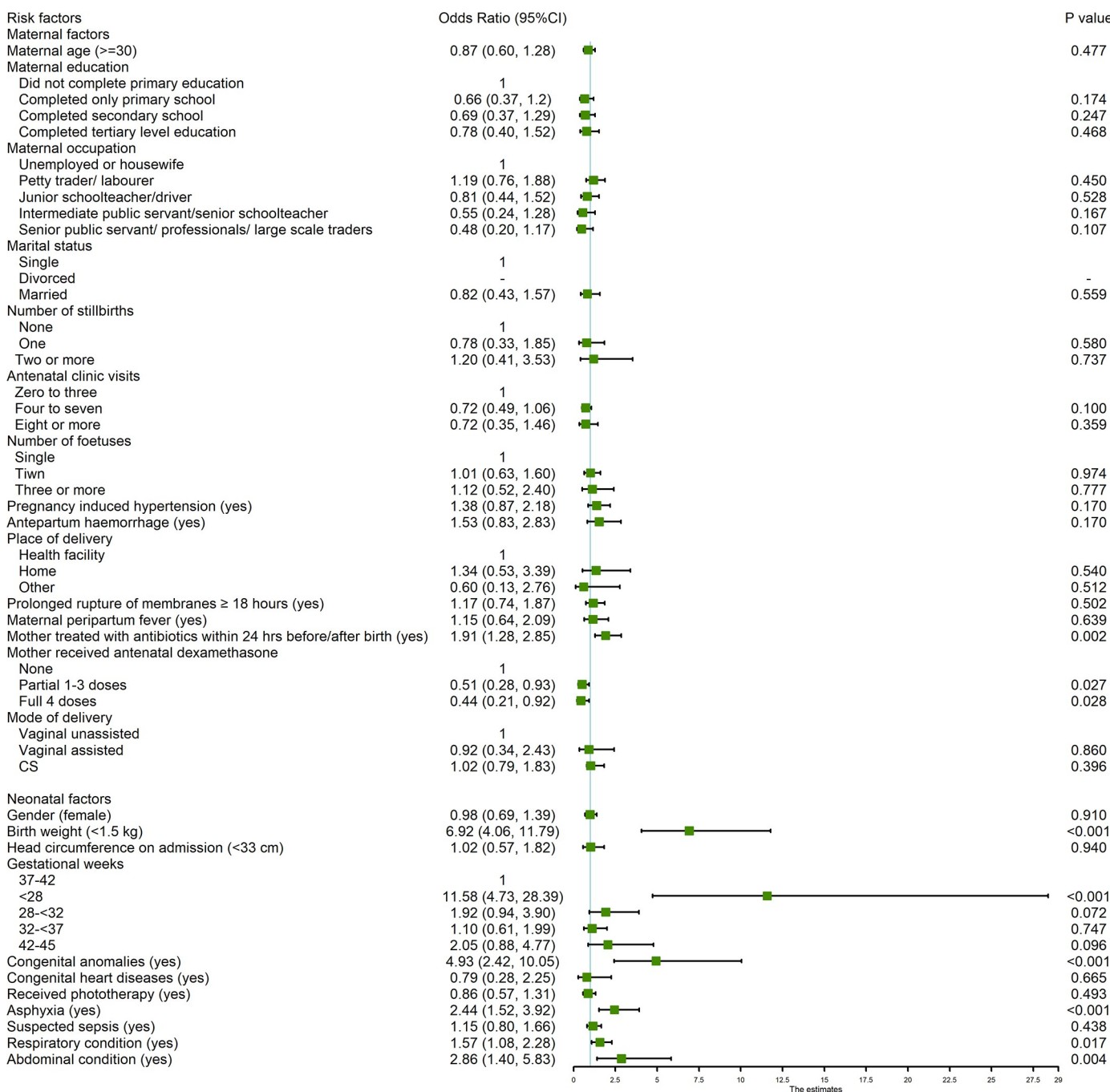

| Risk factors | Odds Ratio (95%CI) | P value |
|---|---|---|
| **Maternal factors** | | |
| Maternal age (>=30) | 0.87 (0.60, 1.28) | 0.477 |
| **Maternal education** | | |
| Did not complete primary education | 1 | |
| Completed only primary school | 0.66 (0.37, 1.2) | 0.174 |
| Completed secondary school | 0.69 (0.37, 1.29) | 0.247 |
| Completed tertiary level education | 0.78 (0.40, 1.52) | 0.468 |
| **Maternal occupation** | | |
| Unemployed or housewife | 1 | |
| Petty trader/ labourer | 1.19 (0.76, 1.88) | 0.450 |
| Junior schoolteacher/driver | 0.81 (0.44, 1.52) | 0.528 |
| Intermediate public servant/senior schoolteacher | 0.55 (0.24, 1.28) | 0.167 |
| Senior public servant/ professionals/ large scale traders | 0.48 (0.20, 1.17) | 0.107 |
| **Marital status** | | |
| Single | 1 | |
| Divorced | - | - |
| Married | 0.82 (0.43, 1.57) | 0.559 |
| **Number of stillbirths** | | |
| None | 1 | |
| One | 0.78 (0.33, 1.85) | 0.580 |
| Two or more | 1.20 (0.41, 3.53) | 0.737 |
| **Antenatal clinic visits** | | |
| Zero to three | 1 | |
| Four to seven | 0.72 (0.49, 1.06) | 0.100 |
| Eight or more | 0.72 (0.35, 1.46) | 0.359 |
| **Number of foetuses** | | |
| Single | 1 | |
| Tiwn | 1.01 (0.63, 1.60) | 0.974 |
| Three or more | 1.12 (0.52, 2.40) | 0.777 |
| Pregnancy induced hypertension (yes) | 1.38 (0.87, 2.18) | 0.170 |
| Antepartum haemorrhage (yes) | 1.53 (0.83, 2.83) | 0.170 |
| **Place of delivery** | | |
| Health facility | 1 | |
| Home | 1.34 (0.53, 3.39) | 0.540 |
| Other | 0.60 (0.13, 2.76) | 0.512 |
| Prolonged rupture of membranes ≥ 18 hours (yes) | 1.17 (0.74, 1.87) | 0.502 |
| Maternal peripartum fever (yes) | 1.15 (0.64, 2.09) | 0.639 |
| Mother treated with antibiotics within 24 hrs before/after birth (yes) | 1.91 (1.28, 2.85) | 0.002 |
| **Mother received antenatal dexamethasone** | | |
| None | 1 | |
| Partial 1-3 doses | 0.51 (0.28, 0.93) | 0.027 |
| Full 4 doses | 0.44 (0.21, 0.92) | 0.028 |
| **Mode of delivery** | | |
| Vaginal unassisted | 1 | |
| Vaginal assisted | 0.92 (0.34, 2.43) | 0.860 |
| CS | 1.02 (0.79, 1.83) | 0.396 |
| | | |
| **Neonatal factors** | | |
| Gender (female) | 0.98 (0.69, 1.39) | 0.910 |
| Birth weight (<1.5 kg) | 6.92 (4.06, 11.79) | <0.001 |
| Head circumference on admission (<33 cm) | 1.02 (0.57, 1.82) | 0.940 |
| **Gestational weeks** | | |
| 37-42 | 1 | |
| <28 | 11.58 (4.73, 28.39) | <0.001 |
| 28-<32 | 1.92 (0.94, 3.90) | 0.072 |
| 32-<37 | 1.10 (0.61, 1.99) | 0.747 |
| 42-45 | 2.05 (0.88, 4.77) | 0.096 |
| Congenital anomalies (yes) | 4.93 (2.42, 10.05) | <0.001 |
| Congenital heart diseases (yes) | 0.79 (0.28, 2.25) | 0.665 |
| Received phototherapy (yes) | 0.86 (0.57, 1.31) | 0.493 |
| Asphyxia (yes) | 2.44 (1.52, 3.92) | <0.001 |
| Suspected sepsis (yes) | 1.15 (0.80, 1.66) | 0.438 |
| Respiratory condition (yes) | 1.57 (1.08, 2.28) | 0.017 |
| Abdominal condition (yes) | 2.86 (1.40, 5.83) | 0.004 |

**Fig 3. Multivariable logistic regression analysis of factors related to mortality.** Missing data have not been included in the analysis. Variables with response rate <90% were omitted from the analysis: HIV status (n = 2024), hepatitis B (n = 1073), syphilis (n = 1387), gestational diabetes (n = 1745), and length on admission (n = 1967). Model performance: Log likelihood = -447.87216, $R^2$ = 0.2792, n = 1,424, P<0.001.

Consistent with data for all neonatal deaths [31, 32], 80% deaths occurred in the early neonatal period (0–6 days). In resource-limited settings, shortages of skilled staff [33] who recognise danger signs and implement appropriate management strategies [34] remain a challenge in neonatal care. In addition, access to essential equipment such as continuous positive airways pressure to support complications such as respiratory distress syndrome [35] is often not

available. Also, as families often incur out of pocket expenses for health care even in public health facilities, this too may hinder quality of care offered to these vulnerable infants [36].

In the multivariable analysis, the highest OR for mortality were for gestation <28 weeks (AOR 11.6, 95% CI 4.7–28.4) and LBW (AOR 6.9, 95% CI 4.1–11.8) and each occurred in about 1 in 5 babies. Estimation of gestation is a challenge in LMICs due to inadequate access to quality antenatal care [37]. Although inferior to an antenatal ultrasound scan, assessment based on date of last menstrual period, the most common method in our study, has been shown to be relatively reliable for the estimation of gestational age in LMICs [38]. Our findings are consistent with the multivariable logistic regression analysis in the prospective studies in Ethiopia, where both gestation <37 weeks and LBW were independently associated with mortality in the multicentre study [21] and gestational age below mean value (<36.6 weeks) in the single centre study [22]. In our analysis and the study by Desalaw *et al.*, although not mutually exclusive, both prematurity and LBW were retained in the multiple regression model suggesting that preterm birth and intra-uterine growth restriction independently increase the risk of mortality. These findings highlight the potential importance of determining the pathogenesis of LBW as optimal management may vary according to underlying causes.

Congenital anomalies occurred in fewer infants (about 1 in 20) but were significantly associated with mortality (AOR 4.9; 95%CI 2.4–10.1). Although not always preventable, congenital anomalies may require referral to tertiary level care with specialised services; therefore, referral pathways and resources to provide appropriate care need to be prioritised in these settings. Abdominal conditions, including NEC, occurred less frequently (3.1% of all infants; 4.8% of infants with gestation < 32 weeks), but were also independently associated with mortality (AOR 2.9, 95%CI 1.4–5.8). NEC is a major disease of preterm infants that hinders the establishment of early nutrition. Optimising newborn feeding is particularly important in contexts where parenteral nutrition is rarely available [39] but there is a lack of pragmatic feeding trials in SSA [40].

Birth asphyxia was diagnosed in about 1 in 4 admissions (1 in 3 of those with gestation ≥35 weeks) and associated with increased mortality (AOR 2.4, 95% CI 1.5–3.9). Our findings are consistent with other tertiary referral NNUs in SSA that receive the most complicated peripartum cases [41, 42] and highlight the need for the review of both obstetric and neonatal procedures based on WHO frameworks for improving the quality of maternal and newborn care [43] particularly at referring facilities.

Most mothers (58.8%) received antibiotics within 24 hours before or after birth. Maternal risk factors for sepsis including prolonged rupture of membranes and peripartum fever were common (16.7% and 9.5% respectively) but antibiotic prescribing exceeded this suggesting that mothers were given antibiotics as a routine despite WHO recommendations [44]. In addition, nearly half of the newborns received prophylactic antibiotics. Suspected sepsis occurred frequently (42% infants) but was not retained in the final multivariable logistic regression model. This likely reflects the poor reliability of clinical diagnosis to identify true sepsis as laboratory investigations were largely unavailable in the Network NNUs. The high exposure to antibiotics amongst both mothers and infants and the fact that maternal exposure to antibiotics was independently associated with infant mortality (AOR 1.9, 95%CI 1.3–2.9) raises concerns about the development of anti-microbial resistance in these settings.

Respiratory conditions were independently associated with mortality (AOR 1.57, 95% CI1.08–2.28) and the timely administration of antenatal dexamethasone to mothers before preterm delivery to prevent or ameliorate neonatal respiratory morbidity was associated with lower mortality. However, additional cost effective interventions to prevent this high burden of respiratory illness in newborns need to be evaluated including increased coverage of maternal influenza immunisation [45, 46].

Neonatal jaundice, requiring treatment with phototherapy, was the commonest neonatal condition (43.6%) and more common in Nigeria than in Kenya. This may reflect differences in newborn care practices and/or regional differences in the frequency of disorders causing neonatal haemolysis such as glucose-6-phosphate dehydrogenase deficiency [47–50]. Although jaundice is often mild, it can progress to kernicterus spectrum disorder, death and long-term severe neurological sequelae in survivors [51–53]. Although we did not find an association between neonatal jaundice and mortality and morbidity among survivors was low, long-term follow-up of survivors may identify motor and cognitive impairments that were not apparent at discharge.

The proportion of mothers <18 years was surprisingly low particularly among the NNUs in Nigeria (0.5%). There are a number of potential reasons for this. Firstly, recruitment of infants <48 hours of age would have excluded some infants delivered at home and presenting later in the postnatal period and this group may have consisted predominantly of younger mothers. Secondly, the out of pocket expenses, particularly at the tertiary NNUs, could have been pro-hibitive for younger mothers, therefore hindering them from accessing specialist neonatal care. An analysis of the 2013 Nigeria Demographic and Health Survey reported that nearly two thirds of women did not deliver in health facilities and the odds of not using health services during delivery were increased among younger women, unmarried women and those from poor households [54]. Access to specialised neonatal care amongst younger and disadvantaged mothers, merits further research.

Maternal mortality was 1.1%, which is much higher than the global estimates of ~0.2% but similar to a multicentre retrospective observational study in tertiary centres in Nigeria (2.1%) [55]. Eclampsia accounted for 5 maternal deaths and pregnancy induced hypertension was the commonest maternal morbidity in the antepartum period. Pregnancy induced hypertension is also associated with adverse neonatal outcomes including preterm/LBW births and stillbirths [56]. A recent meta-analysis of over 800,000 women predominantly from hospital-based urban populations in 24 SSA countries (including Nigeria) found that hypertensive disorders in preg-nancy were common with the highest prevalence rates reported from Central and West African countries [57]. The prevention, identification and management of hypertension should be prioritised in women of childbearing age and in the intrapartum care of women in SSA.

## Strengths and limitations

We consider that prospective data collection, data management systems to identify and address missing data or discrepancies and strong leadership from the co-investigators at all sites enhanced the quality of our data. However, our findings have a number of limitations. Information regarding pregnancies and deliveries were extracted from health records which led to missing data for some maternal health indicators. Most of the admissions in our study were to tertiary level NNUs that are not representative of neonatal care services in Nigeria or Kenya. In settings where health care is not always free at the point of access, the population of mothers who are able to access tertiary referral level neonatal care may be more affluent. Therefore, our data may not be representative of newborns of poorer households and/or admissions to secondary care in SSA. With an observational study design, it is not possible to ascertain cause-effect relationships. Finally, we were not able to establish post-discharge out-comes which are critical for a comprehensive assessment of disease burden in this population.

## Conclusion

This study provides comprehensive multicentre data on the characteristics and short-term out-comes of hospitalised newborns in SSA that will support much needed priority setting for

research and quality improvement in maternal and neonatal care [58]. We have identified a high burden of preventable maternal and neonatal illnesses in Nigeria and Kenya with the highest risk of mortality amongst very preterm and VLBW infants. Our findings emphasize the importance of collaborative work involving maternal and neonatal health clinicians and researchers working in partnership with families to develop strategies to prevent adverse neonatal outcomes that are often closely linked with poor maternal health outcomes [59]. Improved prevention and management of conditions that affect very preterm and VLBW infants deserves to be a health and research priority.

## Supporting information

**S1 Checklist. STROBE statement—checklist of items that should be included in reports of cross-sectional studies.**
(DOCX)

**S1 File. NeoNuNet supplementary tables.**
(DOCX)

**S1 Data. NeoNuNet raw data.** "Demographic, clinical and outcome variables for mothers and newborns".
(XLS)

## Acknowledgments

We would like to thank our colleagues who contributed to the clinical care and collection of data at all the neonatal units in the Network including at the University College Hospital, Ibadan; Lagos University Teaching Hospital; Massey Street Children's Hospital, Lagos; Ahmadu Bello University Teaching Hospital, Zaria and Maitama District Hospital, Abuja in Nigeria. In Kenya they include the Jaramogi Oginga Odinga Teaching and Referral Hospital, Kisumu and The Kilifi County Referral Hospital. We would also like to thank our colleagues at the Nigerian Society of Neonatal Medicine, the Kenya Paediatric Association and the Ministries of Health in Nigeria and Kenya, who provided us with support and advice as we were setting up this study. We thank the mothers for participating in this study with their infants.

Neonatal Nutrition Network members:

Isa Abdulkadir (Ahmadu Bello University, Zaria, Nigeria); Ismaela Abubakar (Liverpool School of Tropical Medicine, Liverpool, UK); Abimbola E Akindolire (College of Medicine, University of Ibadan, Nigeria); Olusegun Akinyinka (College of Medicine, University of Ibadan, Nigeria); Stephen J Allen (Liverpool School of Tropical Medicine, Liverpool, UK); Pauline EA Andang'o (Maseno University, Kenya); Graham Devereux (Liverpool School of Tropical Medicine, Liverpool, UK); Chinyere Ezeaka (Lagos University Teaching Hospital, Nigeria); Beatrice N Ezenwa (Lagos University Teaching Hospital, Nigeria); Iretiola B Fajolu (Lagos University Teaching Hospital, Nigeria); Zainab O Imam (Massey St. Children's Hospital, Lagos, Nigeria); Kevin Mortimer (Liverpool School of Tropical Medicine, Liverpool, UK); Martha K Mwangome (KEMRI Wellcome Trust Research Programme, Kilifi, Kenya); Helen M Nabwera (Liverpool School of Tropical Medicine, Liverpool, UK); Grace M Nalwa (Jaramogi Oginga Odinga Teaching and Referral Hospital, Kisumu, Kenya & Maseno University, Kenya); Walter Otieno (Jaramogi Oginga Odinga Teaching and Referral Hospital, Kisumu, Kenya & Maseno University, Kenya); Alison W Talbert (KEMRI Wellcome Trust Research Programme, Kilifi, Kenya); Nicholas D Embleton (Newcastle University, Newcastle, UK); Olukemi O Tongo (College of Medicine, University of Ibadan, Nigeria); Dominic D Umoru

(Maitama District Hospital, Abuja, Nigeria); Janneke van de Wijgert (University of Liverpool, Liverpool, UK); Melissa Gladstone (University of Liverpool, Liverpool, UK).

Data sharing:

A minimal anonymized data set necessary to replicate the study findings has been shared. Requests for access to further data should be addressed to the corresponding author.

## Author Contributions

**Conceptualization:** Stephen J. Allen.

**Data curation:** Helen M. Nabwera, Dingmei Wang, Pauline E. A. Andang'o, Isa Abdulkadir, Chinyere V. Ezeaka, Beatrice N. Ezenwa, Iretiola B. Fajolu, Zainab O. Imam, Martha K. Mwangome, Dominic D. Umoru, Abimbola E. Akindolire, Walter Otieno, Alison W. Talbert, Ismaela Abubakar, Stephen J. Allen.

**Formal analysis:** Helen M. Nabwera, Dingmei Wang, Stephen J. Allen.

**Funding acquisition:** Helen M. Nabwera, Olukemi O. Tongo, Isa Abdulkadir, Chinyere V. Ezeaka, Iretiola B. Fajolu, Zainab O. Imam, Martha K. Mwangome, Abimbola E. Akindolire, Walter Otieno, Nicholas D. Embleton, Stephen J. Allen.

**Investigation:** Helen M. Nabwera, Olukemi O. Tongo, Pauline E. A. Andang'o, Isa Abdulkadir, Chinyere V. Ezeaka, Iretiola B. Fajolu, Zainab O. Imam, Martha K. Mwangome, Dominic D. Umoru, Abimbola E. Akindolire, Walter Otieno, Grace M. Nalwa, Alison W. Talbert, Ismaela Abubakar, Nicholas D. Embleton, Stephen J. Allen.

**Methodology:** Helen M. Nabwera, Dingmei Wang, Olukemi O. Tongo, Pauline E. A. Andang'o, Isa Abdulkadir, Chinyere V. Ezeaka, Iretiola B. Fajolu, Zainab O. Imam, Martha K. Mwangome, Dominic D. Umoru, Abimbola E. Akindolire, Walter Otieno, Grace M. Nalwa, Alison W. Talbert, Ismaela Abubakar, Nicholas D. Embleton, Stephen J. Allen.

**Project administration:** Olukemi O. Tongo, Pauline E. A. Andang'o, Isa Abdulkadir, Chinyere V. Ezeaka, Iretiola B. Fajolu, Zainab O. Imam, Martha K. Mwangome, Dominic D. Umoru, Abimbola E. Akindolire, Walter Otieno, Grace M. Nalwa, Alison W. Talbert, Stephen J. Allen.

**Resources:** Olukemi O. Tongo, Pauline E. A. Andang'o, Isa Abdulkadir, Chinyere V. Ezeaka, Iretiola B. Fajolu, Zainab O. Imam, Martha K. Mwangome, Dominic D. Umoru, Abimbola E. Akindolire, Walter Otieno, Grace M. Nalwa, Alison W. Talbert, Stephen J. Allen.

**Software:** Olukemi O. Tongo, Isa Abdulkadir, Chinyere V. Ezeaka, Iretiola B. Fajolu, Zainab O. Imam, Martha K. Mwangome, Dominic D. Umoru, Abimbola E. Akindolire, Walter Otieno, Grace M. Nalwa, Alison W. Talbert, Ismaela Abubakar, Stephen J. Allen.

**Supervision:** Helen M. Nabwera, Olukemi O. Tongo, Pauline E. A. Andang'o, Isa Abdulkadir, Chinyere V. Ezeaka, Beatrice N. Ezenwa, Iretiola B. Fajolu, Zainab O. Imam, Martha K. Mwangome, Dominic D. Umoru, Abimbola E. Akindolire, Walter Otieno, Grace M. Nalwa, Alison W. Talbert, Stephen J. Allen.

**Validation:** Helen M. Nabwera, Olukemi O. Tongo, Isa Abdulkadir, Chinyere V. Ezeaka, Iretiola B. Fajolu, Zainab O. Imam, Martha K. Mwangome, Dominic D. Umoru, Abimbola E. Akindolire, Walter Otieno, Grace M. Nalwa, Alison W. Talbert, Ismaela Abubakar, Stephen J. Allen.

**Visualization:** Helen M. Nabwera, Olukemi O. Tongo, Isa Abdulkadir, Chinyere V. Ezeaka, Iretiola B. Fajolu, Zainab O. Imam, Martha K. Mwangome, Dominic D. Umoru, Abimbola E. Akindolire, Walter Otieno, Grace M. Nalwa, Alison W. Talbert, Ismaela Abubakar, Stephen J. Allen.

**Writing – original draft:** Helen M. Nabwera, Dingmei Wang, Stephen J. Allen.

**Writing – review & editing:** Helen M. Nabwera, Dingmei Wang, Olukemi O. Tongo, Pauline E. A. Andang'o, Isa Abdulkadir, Chinyere V. Ezeaka, Beatrice N. Ezenwa, Iretiola B. Fajolu, Zainab O. Imam, Martha K. Mwangome, Dominic D. Umoru, Abimbola E. Akindolire, Alison W. Talbert, Ismaela Abubakar, Nicholas D. Embleton, Stephen J. Allen.

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
