## [Decision Letter · Decision Letter 0]

25 Aug 2020

PONE-D-20-18123

Burden of disease and risk factors for mortality amongst hospitalized newborns in Nigeria and Kenya

PLOS ONE

Dear Dr. Nabwera,

Thank you for submitting your manuscript to PLOS ONE. After careful consideration, we feel that it has merit but does not fully meet PLOS ONE’s publication criteria as it currently stands. Therefore, we invite you to submit a revised version of the manuscript that addresses the points raised during the review process.

We look forward to receiving your revised manuscript.

Kind regards,

Prem S Shekhawat, MD

Academic Editor

PLOS ONE

Additional Editor Comments:

The submission titled "Burden of disease and risk factors for mortality amongst hospitalized newborns in Nigeria and Kenya" is a well-written, simple descriptive study which defines burden of disease in this population from two resource poor countries. The study has merit and needs minor revisions as outlined in comments by two reviewers to make it acceptable for publication.

Journal Requirements:

4. One of the noted authors is a group or consortium [Neonatal Nutrition Network (NeoNuNet)]. In addition to naming the author group, please list the individual authors and affiliations within this group in the acknowledgments section of your manuscript. Please also indicate clearly a lead author for this group along with a contact email address.

5. Your ethics statement must appear in the Methods section of your manuscript. If your ethics statement is written in any section besides the Methods, please move it to the Methods section and delete it from any other section. Please also ensure that your ethics statement is included in your manuscript, as the ethics section of your online submission will not be published alongside your manuscript.

Reviewers' comments:

Reviewer's Responses to Questions

**Comments to the Author**

1. Is the manuscript technically sound, and do the data support the conclusions?

Reviewer #1: Yes

Reviewer #2: Yes

2. Has the statistical analysis been performed appropriately and rigorously? 

Reviewer #1: Yes

Reviewer #2: Yes

3. Have the authors made all data underlying the findings in their manuscript fully available?

Reviewer #1: Yes

Reviewer #2: Yes

4. Is the manuscript presented in an intelligible fashion and written in standard English?

Reviewer #1: Yes

Reviewer #2: Yes

5. Review Comments to the Author

Reviewer #1: Reviewers Comment for manuscript ID: PONE-D-20-18123

1) Abstract: It is adequate, well written and explain according to objective.

2) Introduction: Very well describe the problem statement but comparative data on other middle income countries are missing. Data from other middle income countries can be describe the severity of neonatal health problem. Para 4 can be reduced.

3) Methods:

Study was conducted at different setting (at secondary and tertiary care level) at different countries.

Following question are not clear in methodology

@ Was the admission policy of neonate was same?

@ Who admit the neonates at secondary/ tertiary care level? (Consultant/ Resident)

@ Was the same practice at all NNU?

@ If different persons admitted the neonates, Clinical diagnosis may vary.

@What was the policy for outborn (Home delivered or other facility delivered) neonates?

@Was they admitted with inborn ( same hospital delivered) or at other designated place?

@ What clinical criteria were used to suspect sepsis? You mention only maternal history of premature rupture of membrane and peripartum fever. What about other clinical findings in neonates like lethargy, Sclerema and so on….?

@What about sepsis screen like CRP, micro ESR, IT ratio……?

@What about isolation of Microorganisiam?

@What Respiratory clinical diagnosis were included in respiratory condition? Explain……

@Other than necrotising enterocolitis, what are the other abdominal conditions were included? Explain……..

@How you define birth asphyxia in home delivered neonates?

@What laboratory and radiology investigations were used in diagnosis in study population?

Ethical approval: Taken

Consent of parents: Taken

Statistical analysis: adequate and proper.

Results: Results are properly explain according to objectives.

However, in morbidity and mortality in infants (Table 4)

Jaundice : Serum bilirubin not mention

How many exaggerated physiological, pathological (Rh/ABO incompatibility).

Details about congenital anomalies not mention that leads to death.

How many neonates had life threatening congenital malformations like trachea-esophageal fistula, diaphragmatic hernia or life threatening cardiac malformation like hypoplastic left or right heart syndrome that leads to death in early neonatal period?

Was there any follow up after discharge to say as morbidity?

Discussion:

Very well explain and appropriate.

Conclusion: Appropriate and concise.

Funding: Mention

Competing interest: mention

References: adequate and recent

Tables and figure: Wel explained

Reviewer #2: Comments on the Manuscript: PONE-D-20-18123

Introduction

1. The authors need to provide a detail explanation of the scope of the research

2. The authors need to outline the expected advantages of the research.

Methods

3. Study setting: Why were only these facilities chosen? Could other health facilities in different parts of the countries been included?

4. Are these facilities representative of all neonatal units in Nigeria and Kenya?

5. What was the inclusion and exclusion criteria for the study?

6. Why were newborns aged above 48 hours excluded from the study?

7. Missing data: The authors need to explain the average percentage of missing data for each variable and justify that removing such data does not affect the analysis.

Results

8. Table 3: The authors indicated that 2182 neonates had birth weight recorded. However, percentages are estimated based on 2181 (Section: Newborn admission characteristics). Kindly check and correct the inconsistency.

9. Page 10: 77.9% (81/104) mortality among extremely LBW newborns. However, this number is included in the mortality rate of VLBW newborns (222/472). The authors need to separate these two categories of birthweight to provide a better appreciation of the mortality risk of newborns between 1kg and 1.5kg.

10. Figure 3: Why is the number of categories for some variables (example birthweight) different from table 3? Explain as part of the data analysis why the categories for the regression models different from the descriptive analysis.

6. PLOS authors have the option to publish the peer review history of their article (what does this mean?). If published, this will include your full peer review and any attached files.

Reviewer #1: No

Reviewer #2: **Yes: **Benjamin Atta Owusu

---

## [Author Response · Author response to Decision Letter 0]

26 Oct 2020

Editor’s comments

 We have reviewed and ensured that our manuscript complies with these requirements.

 Thank you. We have included the Tables in the main manuscript.

3. We note that you have indicated that data from this study are available upon request. PLOS only allows data to be available upon request if there are legal or ethical restrictions on sharing data publicly. For information on unacceptable data access restrictions, please see http://journals.plos.org/plosone/s/data-availability#loc-unacceptable-data-access-restrictions

Thank you. We have uploaded the minimal anonymised data set.

4. One of the noted authors is a group or consortium [Neonatal Nutrition Network (NeoNuNet)]. In addition to naming the author group, please list the individual authors and affiliations within this group in the acknowledgments section of your manuscript. Please also indicate clearly a lead author for this group along with a contact email address.

Thank you. We have moved these details to the Acknowledgements section (page 22 lines 5-22). The lead author’s name and contact e-mail address are on page 2.

5. Your ethics statement must appear in the Methods section of your manuscript. If your ethics statement is written in any section besides the Methods, please move it to the Methods section and delete it from any other section. Please also ensure that your ethics statement is included in your manuscript, as the ethics section of your online submission will not be published alongside your manuscript.

We have included the ethics statement in the Methods sections but provided additional details of all the ethics approvals from individual units at the end of the manuscript. We hope this is acceptable. 

Thank you. We have done this.

Reviewer 1:

1. Abstract: It is adequate, well written and explain according to objective.

Thank you.

2. Introduction: Very well described the problem statement but comparative data on other middle income countries are missing. Data from other middle income countries can describe the severity of neonatal health problem. Para 4 can be reduced.

Thank you for this. We have not included data from middle-income countries as the focus of this body of research is to generate evidence for strategies that are relevant to sub-Saharan Africa where majority of countries based on the gross domestic product classification by the International Monetary Fund are classified as low income or lower middle income countries. We therefore felt that it would be relevant to focus on data from this region.

We have deleted paragraph 4.

3) Methods:

Study was conducted at different setting (at secondary and tertiary care level) at different countries.

Following question are not clear in methodology

@ Was the admission policy of neonate was same?

Thank you. All neonatal units admitted both inborn and outborn infants <28 days of age. All neonatal units except one in Kenya had separate rooms/wards for admitting inborn and outborn neonates. We have added these details to the Study setting sub-section (page 7, lines 9-11). 

@ Who admit the neonates at secondary/ tertiary care level? (Consultant/ Resident)

In all the units, the neonates were admitted by intern house officers, medical officers or clinical officers and were reviewed by a consultant neontologist or paediatrician daily during their admission. We have added these details to the Study setting sub-section (page 7, lines 13-16).

@ Was the same practice at all NNU?

Yes, although some NNU’s did not have clinical officers and therefore relied on intern house officers or medical officers.

@ If different persons admitted the neonates, Clinical diagnosis may vary.

Yes, this is the focus on a separate manuscript that we hope to publish soon.

@What was the policy for outborn (Home delivered or other facility delivered) neonates?

As stated above, all neonatal units except one in Kenya had separate rooms/wards for admitting inborn and outborn neonates. We have added these details to the Study setting sub-section (page 7, lines 13-16). 

@Was they admitted with inborn ( same hospital delivered) or at other designated place?

The infants who were outborn were admitted to separate rooms/wards for all neonatal units except one in Kenya. We have added these details to the Study setting sub-section (page 7, lines 13-16). 

@ What clinical criteria were used to suspect sepsis? You mention only maternal history of premature rupture of membrane and peripartum fever. What about other clinical findings in neonates like lethargy, Sclerema and so on….?

We had no set criteria for sepsis as we were keen not to be prescriptive but to let individual neonatal units use their own criteria for diagnosing sepsis. The case report form that we developed through consensus among all the clinical leads for the seven neonatal units and other newborn health care providers from the Nigeria Society of Neonatal Medicine and the Kenya Paediatric Association, is available on our Neonatal Nutrition Network website as stated in this manuscript. We collected this data systematically and as mentioned are analyzing this for a separate manuscript (page 8, lines 17-19). In summary, there was a lot of variability in the criteria used for diagnosing sepsis that focused more on infant clinical signs, most frequently temperature instability and tachypnoea.

@What about sepsis screen like CRP, micro ESR, IT ratio……?

As stated above, these analyses are in a separate manuscript. Briefly, laboratory investigations were rarely used to diagnose sepsis. The case report form that we developed through consensus among all the clinical leads for the seven neonatal units and other newborn health care providers from the Nigeria Society of Neonatal Medicine and the Kenya Paediatric Association, is available on our Neonatal Nutrition Network website as stated in this manuscript. 

@What about isolation of Microorganism?

As stated above, these details are provided in a separate manuscript that is in preparation. Only one of the units had consistently reliable microbiology services therefore isolation of microorganisms from blood, cerebrospinal fluid or urine in the context of a diagnosis of sepsis was uncommon. The case report form that we developed through consensus among all the clinical leads for the seven neonatal units and other newborn health care providers from the Nigeria Society of Neonatal Medicine and the Kenya Paediatric Association, is available on our Neonatal Nutrition Network website as stated in this manuscript. 

@What Respiratory clinical diagnosis were included in respiratory condition? Explain……

As stated above, these details are provided in a related manuscript in preparation. Briefly, the most used criteria were respiratory distress, being preterm infant< 37 weeks gestation, history of meconium stained aspiration and maternal risk factors for sepsis. The case report form that we developed through consensus among all the clinical leads for the seven neonatal units and other newborn health care providers from the Nigeria Society of Neonatal Medicine and the Kenya Paediatric Association, is available on our Neonatal Nutrition Network website as stated in this manuscript. 

@Other than necrotising enterocolitis, what are the other abdominal conditions were included? Explain……..

Other abdominal conditions include septic ileus, dysmotility and others including congenital anomalies including jejunal, duodenal, rectal atresia, omphalocele/gastroschisis and bilateral obstructed inguinal hernia. The case report form that we developed through consensus among all the clinical leads for the seven neonatal units and other newborn health care providers from the Nigeria Society of Neonatal Medicine and the Kenya Paediatric Association, is available on our Neonatal Nutrition Network website as stated in this manuscript. 

@How you define birth asphyxia in home delivered neonates?

Birth asphyxia in home delivered neonates was defined based on their condition at the time of admission. This was mainly based in clinical criteria, most commonly, evidence of encephalopathy, evidence of multiorgan dysfunction and exclusion of other aetiologies. The case report form that we developed through consensus among all the clinical leads for the seven neonatal units and other newborn health care providers from the Nigeria Society of Neonatal Medicine and the Kenya Paediatric Association, is available on our Neonatal Nutrition Network website as stated in this manuscript. 

@What laboratory and radiology investigations were used in diagnosis in study population?

As stated above, these details will be provided in a related manuscript. Briefly, the use of either radiology or laboratory was uncommon across all the 7 NNUs. 

Ethical approval: Taken

Consent of parents: Taken

Statistical analysis: adequate and proper.

Thank you.

Results: Results are properly explained according to objectives.

However, in morbidity and mortality in infants (Table 4)

Jaundice : Serum bilirubin not mentioned

As serum bilirubin was not done routinely across the NNUs, jaundice was defined as infant treated with phototherapy. We have stated this on page 12, lines 4-5.

How many exaggerated physiological, pathological (Rh/ABO incompatibility).

Thank you. Unfortunately, we do not have these details and laboratory investigations were limited.

Details about congenital anomalies not mentioned that leads to death.

Of the 128 infants who were diagnosed with congenital anomalies, 35 died during admission. These included: 10 with congenital gastrointestinal abnormalities (1 with jejunal atresia, 1 with duodenal atresia, 1 with a gastric mass, 4 with gastroschisis, 3 with omphalocele), 6 with congenital central nervous system abnormalities (2 with congenital hydrocephalus, 2 with lumbar myelomeningocele, 1 with frontal encephalocele, 1 with microcephaly), 5 cartilage and limb abnormalities (1 with Achondroplasia, 1 with chrondrodysplasia, 1 congenital amputation of the right foot, 1 with genu recarvatum, 1 with congenital talipes), 4 with congenital heart disease (both cyanotic and acyanotic), 2 with facial abnormalities ( 1 with cleft lip and palate, 1 with syngnathia) and 8 with other anomalies (1 with ambiguous genitalia, 1 Dysmorphic features (Down’s syndrome), 1 with Prune Belly syndrome, 1 with hydrops fetalis, 2 with low set ears, 1 with multiple anomalies and 1 with posterior urethral valves). We have included these details in the Results (page 12, lines 8-16)

How many neonates had life threatening congenital malformations like trachea-esophageal fistula, diaphragmatic hernia or life-threatening cardiac malformation like hypoplastic left or right heart syndrome that leads to death in early neonatal period?

As shown in Table 4, only 5.6% (128) of neonates who were admitted to the 7 NNUs had congenital anomalies. We have summarized the details of the congenital anomalies that were associated with neonatal mortality across the 7 NNU above and in the manuscript. 

Was there any follow up after discharge to say as morbidity?

No, in this study we did not collect post discharge outcome data on the infants but plan to do this in the future as we appreciate that the health, growth and developmental outcomes of these infants post discharge are important indicators of the quality of maternal and newborn care. 

Discussion:

Very well explain and appropriate.

Thank you.

Conclusion: Appropriate and concise.

Thank you.

Funding: Mention

Competing interest: mentioned

Thank you.

References: adequate and recent

Thank you.

Tables and figure: Well explained

Thank you.

Reviewer 2: 

Introduction

1. The authors need to provide a detail explanation of the scope of the research.

Thank you. This is paper contains the details of the first phase of an ambitious project- the Neonatal Nutrition Network project(https://www.lstmed.ac.uk/nnu) that seeks to evaluate nutrition interventions for preterm/ low birth weight infants in sub-Saharan Africa as a key strategy for improving their survival and long term health, growth and neurodevelopmental outcomes. In this paper we describe the burden of neonatal disease in 7 neonatal units in Nigeria and Kenya. These data will form the basis for future intervention studies in these neonatal units. We have now emphasized this more in the Introduction to ensure that readers get a clearer view of the scope of research (page 4 line 1 to page 6, line 5). 

2. The authors need to outline the expected advantages of the research.

Thank you. Reliable routine clinical data is a prerequisite to the prioritization, design and implementation of interventions. These data are often lacking in sub-Saharan Africa making difficult for neonatal units to benchmark and prioritise care in the context of limited resources. Our research will therefore provide a platform for shared learning in addition to providing key baseline data on neonatal morbidity and mortality that will be used to inform the design and evaluation of interventions to address the burden. We have emphasized this in the Introduction (page 5, line 25 to page 6, line 5). 

Methods

3. Study setting: Why were only these facilities chosen? Could other health facilities in different parts of the countries been included?

Thank you. In Nigeria, the UK co-investigators engaged with the Nigeria Society of Neonatal Medicine who led in the selection of these facilities aiming to incorporate neonatal units from both the northern and southern parts of the country. In Kenya, the facilities were chosen based on previous collaborative partnerships that the UK co-investigators had with child health researchers aiming to include neonatal units that provide different levels of care i.e. tertiary and district level in different regions of the country. We have included these details in the Methods section (page 7, line 1-9) 

4. Are these facilities representative of all neonatal units in Nigeria and Kenya?

No, these facilities are not representative of neonatal units in Nigeria and Kenya. We have emphasized this under the Strengths and Limitations section (page 19, lines 20-21).

5. What was the inclusion and exclusion criteria for the study?

All newborns aged <48 hours who were admitted to each of NNU’s over the 6-month period that fell between August 2018 and May 2019 for all NNUs (depending on the time when they obtained ethics approval). Infants were excluded if they were ≥ 48 hours of age at admission. We have clarified these criteria on page 8, lines 7-13.

6. Why were newborns aged above 48 hours excluded from the study?

We wanted to optimize recall of information on feeding practices from birth. In addition, infants aged 48 hours and above are generally admitted to paediatric wards rather than neonatal units. Therefore, these were not included in our Network.

7. Missing data: The authors need to explain the average percentage of missing data for each variable and justify that removing such data does not affect the analysis.

This was an observational study, which was based on the real-world data. Except for five variables with higher percentage of missing data (>=10%, HIV status, hepatitis B, syphilis, gestational diabetes, and length on admission), the average percentage of missing data for variables ranged from 0-6.3%. The five variables with a high percentage were not included in the multivariable logistic regression analysis. We have provided total number of mothers/infants with available data for each variable and the total number of participants to enable readers to derive the details of the missing data. We have added these details to the Data analysis subsection (page 9, lines 8-11). 

Results

8. Table 3: The authors indicated that 2182 neonates had birth weight recorded. However, percentages are estimated based on 2181 (Section: Newborn admission characteristics). Kindly check and correct the inconsistency.

Thank you. We have checked that it was 2182 neonates had birth weight and percentages also were estimated based on 2182; corrected on page 11 line 14.

9. Page 10: 77.9% (81/104) mortality among extremely LBW newborns. However, this number is included in the mortality rate of VLBW newborns (222/472). The authors need to separate these two categories of birthweight to provide a better appreciation of the mortality risk of newborns between 1kg and 1.5kg.

Thanks for your advice. We have added a category of birthweight <1kg with “Final outcome among infants birth weight <1.0kg” in Table 4.

10. Figure 3: Why is the number of categories for some variables (example birthweight) different from table 3? Explain as part of the data analysis why the categories for the regression models different from the descriptive analysis.

Figure 3 is a multivariable logistic regression analysis that includes all available data but, to avoid potential bias produced by missing data, the variables with low response rate (<90%, <2036: HIV status (n=2024), hepatitis B (n=1073), syphilis (n=1387), gestational diabetes (n=1745), and length on admission (n=1967)) were not included in this model. The model excluded participants with missing data automatically. As a result, 1,424 participants with complete data were included in the multivariable logistic regression model.

Additionally, the number of neonates in Figure 1 (birthweight) and Figure 2 (gestation) have minor differences from Table 3 because there were some missing outcome date data, resulting in exclusions in the KM survival estimates.

---

## [Decision Letter · Decision Letter 1]

3 Dec 2020

Burden of disease and risk factors for mortality amongst hospitalized newborns in Nigeria and Kenya

PONE-D-20-18123R1

Dear Dr. Nabwera,

We’re pleased to inform you that your manuscript has been judged scientifically suitable for publication and will be formally accepted for publication once it meets all outstanding technical requirements.

Kind regards,

Prem Singh Shekhawat, MD

Academic Editor

PLOS ONE

Reviewers' comments:

Reviewer's Responses to Questions

**Comments to the Author**

1. If the authors have adequately addressed your comments raised in a previous round of review and you feel that this manuscript is now acceptable for publication, you may indicate that here to bypass the “Comments to the Author” section, enter your conflict of interest statement in the “Confidential to Editor” section, and submit your "Accept" recommendation.

Reviewer #1: All comments have been addressed

Reviewer #2: All comments have been addressed

2. Is the manuscript technically sound, and do the data support the conclusions?

Reviewer #1: Yes

Reviewer #2: Yes

3. Has the statistical analysis been performed appropriately and rigorously? 

Reviewer #1: Yes

Reviewer #2: Yes

4. Have the authors made all data underlying the findings in their manuscript fully available?

Reviewer #1: Yes

Reviewer #2: Yes

5. Is the manuscript presented in an intelligible fashion and written in standard English?

Reviewer #1: Yes

Reviewer #2: Yes

6. Review Comments to the Author

Reviewer #1: Abstract: It is adequate, well written and explain according to objective.

Introduction: Correction done according to comments but still I feel, though this study focus to generate the evidences from sub-Saharahan Africa, comparative data of other middle income countries will is important as a readers view.

Methods: Adequate and clear the idea.

Ethical approval: Taken

Consent of parents: Taken

Statistical analysis: adequate and proper.

Results: Results are properly explained according to objectives.

Discussion:

Very well explain and appropriate.

Conclusion: Appropriate and concise.

Funding: Mention

Competing interest: mention

References: adequate and recent

Tables and figure: Well explained

Reviewer #2: (No Response)

7. PLOS authors have the option to publish the peer review history of their article (what does this mean?). If published, this will include your full peer review and any attached files.

Reviewer #1: **Yes: **Rajkumar Motiram Meshram

Reviewer #2: **Yes: **Benjamin Atta Owusu

---

## [Editor Report · Acceptance letter]

5 Jan 2021

PONE-D-20-18123R1 

Burden of disease and risk factors for mortality amongst hospitalized newborns in Nigeria and Kenya 

Dear Dr. Nabwera:

I'm pleased to inform you that your manuscript has been deemed suitable for publication in PLOS ONE. Congratulations! Your manuscript is now with our production department. 

Kind regards, 

on behalf of

Dr. Prem Singh Shekhawat 

Academic Editor

PLOS ONE